# Screening drug effects in patient-derived cancer cells links organoid responses to genome alterations

Julia Jabs[1,2,3] (iD), Franziska M Zickgraf[4,5] (iD), Jeongbin Park[1,3] (iD), Steve Wagner[4,5], Xiaoqi Jiang[6], Katharina Jechow[1,2], Kortine Kleinheinz[1,2,3] (iD), Umut H Toprak[1], Marc A Schneider[7] (iD), Michael Meister[7], Saskia Spaich[8], Marc Sütterlin[8] (iD), Matthias Schlesner[1] (iD), Andreas Trumpp[4,5,9], Martin Sprick[4,5,9] (iD), Roland Eils[1,2,3,10,*] (iD) & Christian Conrad[1,2,**] (iD)

## Abstract

Cancer drug screening in patient-derived cells holds great promise for personalized oncology and drug discovery but lacks standardization. Whether cells are cultured as conventional monolayer or advanced, matrix-dependent organoid cultures influences drug effects and thereby drug selection and clinical success. To precisely compare drug profiles in differently cultured primary cells, we developed *DeathPro*, an automated microscopy-based assay to resolve drug-induced cell death and proliferation inhibition. Using *DeathPro*, we screened cells from ovarian cancer patients in monolayer or organoid culture with clinically relevant drugs. Drug-induced growth arrest and efficacy of cytostatic drugs differed between the two culture systems. Interestingly, drug effects in organoids were more diverse and had lower therapeutic potential. Genomic analysis revealed novel links between drug sensitivity and DNA repair deficiency in organoids that were undetectable in monolayers. Thus, our results highlight the dependency of cytostatic drugs and pharmacogenomic associations on culture systems, and guide culture selection for drug tests.

**Keywords** cancer organoids; confocal microscopy; high-throughput screening; personalized drug screen; pharmacogenomics
**Subject Categories** Cancer; Methods & Resources; Pharmacology & Drug Discovery
**Mol Syst Biol. (2017) 13: 955**

## Introduction

Cell-based assays are a key tool in basic research and drug discovery, and are increasingly used in personalized oncology. In the last years, numerous anticancer therapeutics developed from standard cell line screens in conventional 2D culture failed in clinical studies (Horvath *et al*, 2016). As a result, standard treatment and overall survival of advanced cancers like ovarian cancer (OC) has not changed for decades (Bowtell *et al*, 2015). To allow personalized therapy and improve drug development, new patient-derived models such as organoids (Gao *et al*, 2014; Van de Wetering *et al*, 2015; Schütte *et al*, 2017) and patient-derived xenografts (Alkema *et al*, 2015; Gao *et al*, 2015; Bruna *et al*, 2016) that recapitulate the heterogeneity and intrinsic drug sensitivity of the original tumour have started to replace the popular cancer cell lines. Patient-derived organoids may be grown as 3D cultures on hydrogels like Matrigel that mimic the extracellular matrix. Compared to 2D cell cultures, they have emerged as near-physiological models reflecting the gene expression, differentiation and structure of the primary tissue (Fatehullah *et al*, 2016). Nevertheless, due to increased workload, higher costs and the current lack of 3D assay methods, most drug screens are still performed in less physiological 2D cultures (Edmondson *et al*, 2014). Initial studies in ovarian cancer showed that cells cultured as cell aggregates are less sensitive to drugs than in monolayer culture (Loessner *et al*, 2010; Lee *et al*, 2013). The culture format thus shapes cellular drug responses and defines the translational power of a drug assay. However, this dependency cannot be studied in detail with widely used, unspecific viability assays that measure metabolic activity or cellular ATP as surrogate markers. Such assays show limited reproducibility and do not

1 Division of Theoretical Bioinformatics, German Cancer Research Center (DKFZ), Heidelberg, Germany
2 Center for Quantitative Analysis of Molecular and Cellular Biosystems (BioQuant), University of Heidelberg, Heidelberg, Germany
3 Department for Bioinformatics and Functional Genomics, Institute for Pharmacy and Molecular Biotechnology (IPMB) and BioQuant, Heidelberg University, Heidelberg, Germany
4 Heidelberg Institute for Stem Cell Technology and Experimental Medicine (HI-STEM) gGmbH, Heidelberg, Germany
5 Division of Stem Cells and Cancer, German Cancer Research Center (DKFZ), Heidelberg, Germany
6 Division of Biostatistics, German Cancer Research Center (DKFZ), Heidelberg, Germany
7 Thoraxklinik at Heidelberg University Hospital, Member of the German Center for Lung Research (DZL), Heidelberg, Germany
8 Department of Gynaecology and Obstetrics, University Medical Centre Mannheim, University of Heidelberg, Mannheim, Germany
9 German Cancer Consortium, Heidelberg, Germany
10 Heidelberg Center for Personalized Oncology, DKFZ-HIPO, DKFZ, Heidelberg, Germany
*Corresponding author. Tel: +49 6221 42 3600; E-mail: r.eils@dkfz.de
**Corresponding author. Tel: +49 6221 54 51304; E-mail: c.conrad@dkfz.de

resolve actual drug effects of high therapeutic interest such as cell death and growth arrest (Haibe-Kains *et al*, 2013; Van de Wetering *et al*, 2015). Instead, recent advances in automated microscopy enable more sophisticated assays that can deconvolve drug effects in different culture formats.

Here, we systematically compare drug effects in organoid and standard 2D culture using *DeathPro*, a confocal microscopy-based assay and image processing workflow to simultaneously study cell death and growth arrest in patient-derived material over time. Using *DeathPro*, we screened cells from nine high-grade serous OC patients with clinically relevant drugs and found that growth arrest and the efficacy of cytostatic drugs notably depend on the culture type. Remarkably, patient-specific genomic alterations correlated with drug effects observed in organoids, but not in 2D cell monolayers. Hence, combining refined assays like *DeathPro* with advanced models like cancer organoids could enhance drug screening in the context of personalized oncology and pharmacogenomics.

## Results

### Deconvolving drug-induced cell death and proliferation inhibition

To resolve drug effects in patient cells and organoids, we developed an automated live cell assay and quantification workflow, which deconvolves drug-induced death and proliferation inhibition over time (*DeathPro*; Fig 1). To this end, cells were stained with Hoechst and counterstained with propidium iodide (PI) for dead cells and analysed at consecutive time points by confocal microscopy. To accurately quantify cell growth for each condition (Hafner *et al*, 2016), cells were imaged at the start and end of the drug treatment at the same position (Fig 1A). For high-throughput image analysis, we built an adaptable visual programming workflow that encompasses adaptive sequential thresholding and outlier filtering strategies to cope with heterogeneous cell morphologies and dye intensities. In the workflow, total areas covered by dead cells (PI-stained) and all cells (Hoechst or PI-stained) were determined from confocal images and used to calculate LD50 values and area under curve values for cell death (AUCd) and proliferation inhibition (AUCpi; Fig 1B).

The *DeathPro* assay and workflow reliably resolved carboplatin-induced cell death and proliferation inhibition in OC organoids generated by culturing patient-derived cells on Matrigel (Fig 1). In addition, we performed pilot drug screens in OC patient cells from mouse xenografts and in 2D co-cultures with fibroblasts to validate our *DeathPro* concept in other common personalized cancer models (Fig EV1A–D). Moreover, we resolved drug effects in lung cancer organoids to verify that the *DeathPro* workflow can be applied to patient cells from different cancer entities (Fig EV1E and F).

By using live cell dyes, patient cells or organoids can be directly used for screening and do not have to be genetically modified to express fluorescent proteins. To exclude the possibility that either dye alters cell behaviour, we tested their effect on OC organoids. Hoechst and PI did not affect organoid growth but increased cell death (Fig EV2A and B), which is accounted for in AUCd measurements by normalization to the untreated control (Fig 1B).

Additionally, cytotoxic effects induced by 11 drugs correlated well between long-term and short-term stained organoids (Pearson correlation 0.81–0.95) indicating that both dyes do not interfere with drug-induced cell death measured by *DeathPro* (Fig EV2C and D). Imaging OC12 organoids only at the end or additionally at the beginning of the drug treatment did neither alter organoid growth nor cell death (Fig EV2A). To achieve low phototoxicity and high throughput of *DeathPro*, we chose to acquire confocal images at low resolution and to analyse 2D image projections. To validate that this coarse procedure captures complex 3D phenotypes, we experimentally compared the *DeathPro* strategy to "slice-wise" analysis of confocal image stacks. In the tested conditions, we detected similar cell death ratios with both approaches (Appendix Fig S1). Thus, the *DeathPro* imaging strategy can be used to efficiently determine drug effects in screens but at the cost of a potential bias which we cannot exclude for all conditions.

### Drug-induced growth arrest in ovarian cancer patient cells is culture-dependent

To systematically assess the influence of extracellular matrix on patient cell responses, we used the *DeathPro* assay to screen patient-derived OC cell lines (PDCLs) in standard 2D culture or as cancer organoids. PDCLs were established from metastatic serous ovarian cancers, maintained in 2D culture and seeded on Matrigel to generate "cancer organoids" (FIGO stage IIIc-IV, Table EV1, Fig 2A). Additionally, we included human ovarian surface epithelial cells (HOSEpiC) to assess potential side-effects such as cytotoxicity in normal cells. Seeded on Matrigel, HOSEpiC developed into spheres whereas PDCLs formed morphologically diverse "cancer organoids" (Fig 2B) that expressed the tumour markers CA-125 and WT1 (Appendix Fig S2).

Ovarian cancer organoids or 2D cultured PDCLs were screened twice for 22 drugs or drug combinations (Table EV2) currently used or under investigation for treatment of OC. LD50 and cell death (AUCd) values were highly reproducible across all drugs and patients in 2D and organoid culture (Pearson correlation 0.86–0.97, Fig EV3A), whereas growth arrest (AUCpi) showed slightly lower correlation (Pearson correlation 0.67–0.76, Fig EV3B).

Based on the *DeathPro* results, we compared all drug effects determined in OC patient cells between 2D culture and 3D culture (Fig 2C). In both screens, drugs induced more growth arrest than cell death (Fig 2D). Due to low drug-induced cell death, LD50 values could not be determined in 20–30% of all conditions (Fig EV4A). After 72-h drug treatment, cell death was slightly lower in organoids than that in 2D cultures (Figs 2C and EV4B). Surprisingly, death upon drug treatment strongly correlated in 2D and 3D culture whereas drug-induced growth arrest varied greatly with culture type (Fig 2D, Pearson correlation 0.85 vs. 0.475). Since drug-induced cell death was growth-dependent and organoids grew slowly compared to cells in 2D culture (Fig EV3C), we measured organoid responses a second time after drug removal in 3D (Fig 2E, Appendix Fig S3A). After washout, drug effects increased in most patient organoids (Appendix Fig S3A and B) as they either intensify with time or continue to be induced by residual compounds in Matrigel. Still, cytotoxicity levels resembled those in 2D culture (Fig 2F, Pearson correlation 0.755). Likewise, LD50s measured in 3D culture before and after drug removal highly correlated with

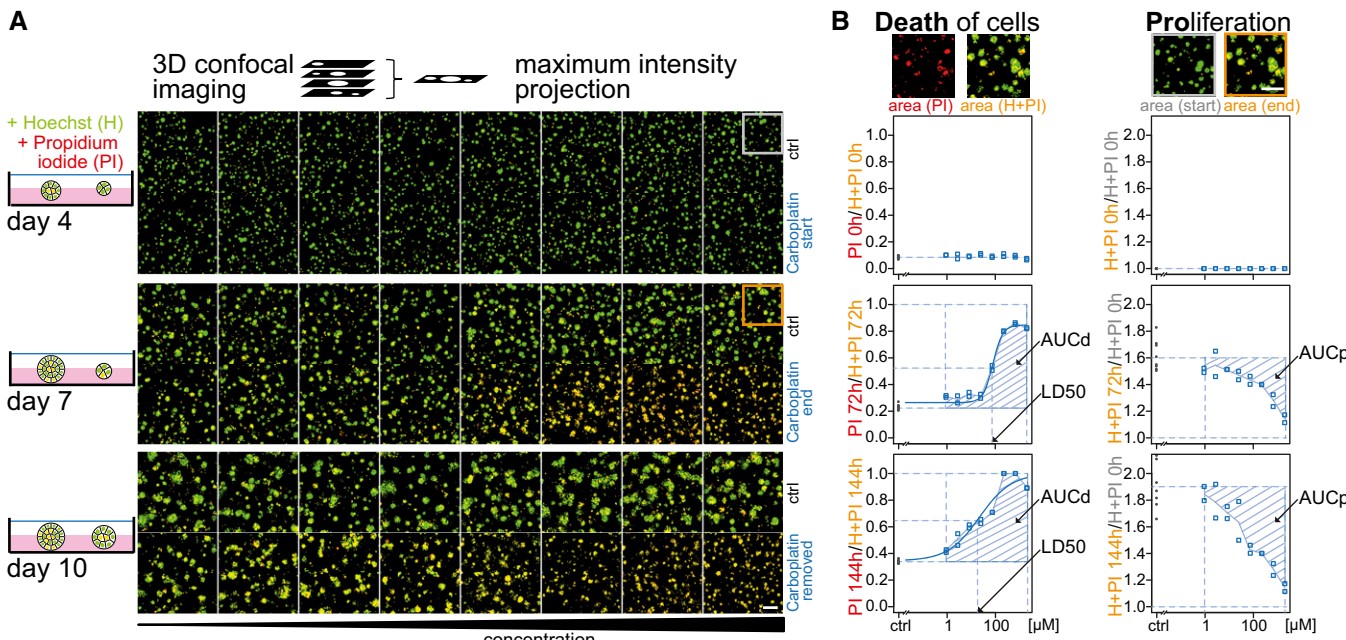

**Figure 1.  Drug-induced cell death and proliferation inhibition can be quantified from serial confocal images.**

A    Schematic overview of drug testing in organoid culture with the *DeathPro* assay. Cells are grown on Matrigel for 4 days, stained with Hoechst (H) and propidium iodide (PI) and imaged at day 4, day 7 and day 10. Image gallery exemplifies OC12 organoid growth and cell death at start (day 4) and end of carboplatin treatment (day 7) and after carboplatin removal (day 10) using eight carboplatin concentrations or drug-free medium (ctrl). Confocal images are reduced to maximum intensity projections, and binary images of merged Hoechst (green) and PI (red) channel are shown.

B    Image analysis for the *DeathPro* assay is based on area measurements in Hoechst and PI channels, and calculation of LD50, AUCd and AUCpi values to describe cell death and growth arrest. Drug response curve fitting and AUC values are illustrated for OC12 at time points depicted in (A).

Data information: Grey and orange boxes in (A) correspond to the magnifications in (B). Scale bar is 200 μm.

---

LD50s in 2D culture (Fig EV4C and D, Pearson correlation 0.872, 0.822). In contrast, growth inhibition again differed after drug removal (Fig 2F, Pearson correlation 0.525). Overall, growth arrest was the major drug effect in OC cells and was culture type-dependent, whereas cell death was similar between culture types.

**Efficacy of cytostatic drugs depends on culture type**

As the culture type affected growth arrest, the efficacy of cytostatic drugs that do not induce cell death should be culture type-dependent as well. Thus, we compared the efficacy of drugs in our panel by summarizing drug response parameters (LD50, AUCd and AUCpi) into a single efficacy measure, and clustering drugs on this basis. In 2D and 3D culture, three clusters arose based on differential cytotoxicity: (i) drugs effectively inducing cell death and growth inhibition (red cluster), (ii) medium cytotoxic drugs (yellow cluster), and (iii) ineffective drugs (blue cluster, Fig 3A and B). Clustering revealed that the most effective treatments (red cluster, Fig 3A and B) in both screens comprised belinostat, BKM120, the first-line therapeutic carboplatin and all combinations thereof. Paclitaxel, which forms part of the current first-line therapy for OC, was not among the most effective treatments tested due to its low toxicity in most patient cells (Fig 2C and E). Moreover, its combination with carboplatin performed no better than carboplatin alone in 2D and 3D (Appendix Fig S3C and D).

A fourth drug efficacy cluster appearing in 3D, but not in 2D culture, included four drugs that induced strong growth arrest but low cell death (green cluster, Fig 3A and B). All four drugs, ICG-001, temsirolimus, AZD5363 and AZD2014, target proliferation pathways and were more effective in 3D culture. To differentiate between these and other drugs, we divided our panel into "cytostatic drugs" inhibiting kinases or other effectors of proliferation pathways and "cytotoxic drugs" causing DNA damage, DNA methylation changes or mitotic failure.

The effects of four specific drugs and one drug combination were significantly altered in 2D compared to 3D before and after drug removal (Fig 3C and D): sarcoma (SRC) kinase inhibitor dasatinib induced significantly lower cell death and growth arrest in OC organoids than in monolayer patient cells (Fig 3E). In combination with carboplatin and paclitaxel, growth arrest in organoids was still lower than that in 2D. In contrast to dasatinib, the mTOR inhibitors temsirolimus and AZD2014 inhibited cell growth in organoids more strongly than in 2D culture. Azacytidine was the only cytotoxic drug that induced lower growth arrest in organoids than in cells in 2D. As azacytidine induced comparable cell death in 2D and 3D (Fig 2C and E), its overall efficacy was similar in 2D and 3D culture (yellow, medium cytotoxic cluster Fig 3A). In both screens, we found that belinostat, BKM120 and carboplatin were the most potent drugs and that the efficacy of the cytostatic drugs dasatinib, temsirolimus and AZD2014 depended on culture type.

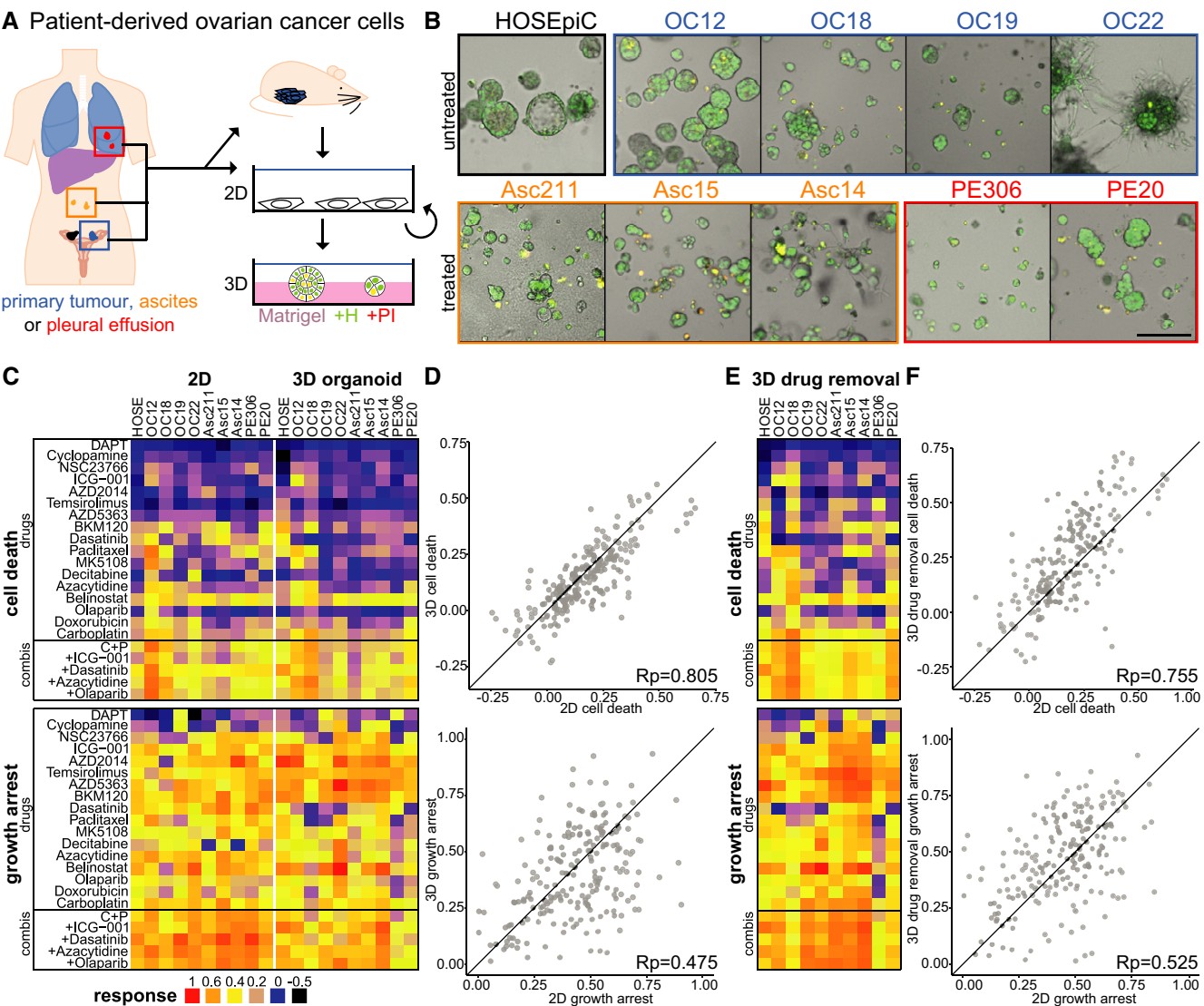

**Figure 2. Culture type shapes drug-induced growth arrest in ovarian cancer patient cells.**

A   Simplified overview of generation and cultivation of patient-derived ovarian cancer cell lines (PDCLs) from different sites (OC: primary tumour, Asc: ascites, PE: pleural effusion). Patient material was taken directly into 2D culture (Asc211, PE306) or amplified by xenografting into mice. PDCLs are maintained in 2D culture but can be grown as ovarian cancer organoids on Matrigel.

B   Morphology of ovarian cancer organoids and normal ovarian epithelial cells (HOSEpiC) on Matrigel 7 days after seeding. Green (Hoechst) and red (PI) channels are merged.

C   Drug responses (cell death: AUCd, growth arrest: AUCpi) measured with *DeathPro* assay after 72-h drug treatment in patient cells cultured as monolayers (2D) or ovarian cancer organoids (3D).

D   Comparison of drug-induced cell death (AUCd) and growth arrest (AUCpi) in 2D vs. 3D.

E   Drug responses measured in ovarian cancer organoids (3D) after 72-h drug treatment followed by 72-h drug removal.

F   Comparison of drug-induced cell death and growth arrest in 2D vs. 3D after drug removal.

Data information: All values shown are means of two independent biological replicates. HOSE, HOSEpiC; Rp, Pearson correlation coefficient; C + P, carboplatin + paclitaxel.

As drug responses in cancer cells can be influenced by stromal cells, we investigated how drug effects change when OC cells in 2D are co-cultured with primary ovary or lung fibroblasts that model cell interactions in the primary tumour or in lung metastasis, respectively. We tested four PDCLs against five OC drugs and found that drug-induced cytotoxicity and growth arrest were highly correlated between both co-culture types (Rp = 0.96, 0.66, Fig EV5A and B). Drug responses in co-cultures resembled 2D culture effects closer than 3D culture responses (Rp = 0.64, 0.68 vs. −0.16, −0.05 in 2D and 3D, Fig EV5A and B). This points to a high influence of the culture format on drug responses that even persists when the model is expanded by including other cell types.

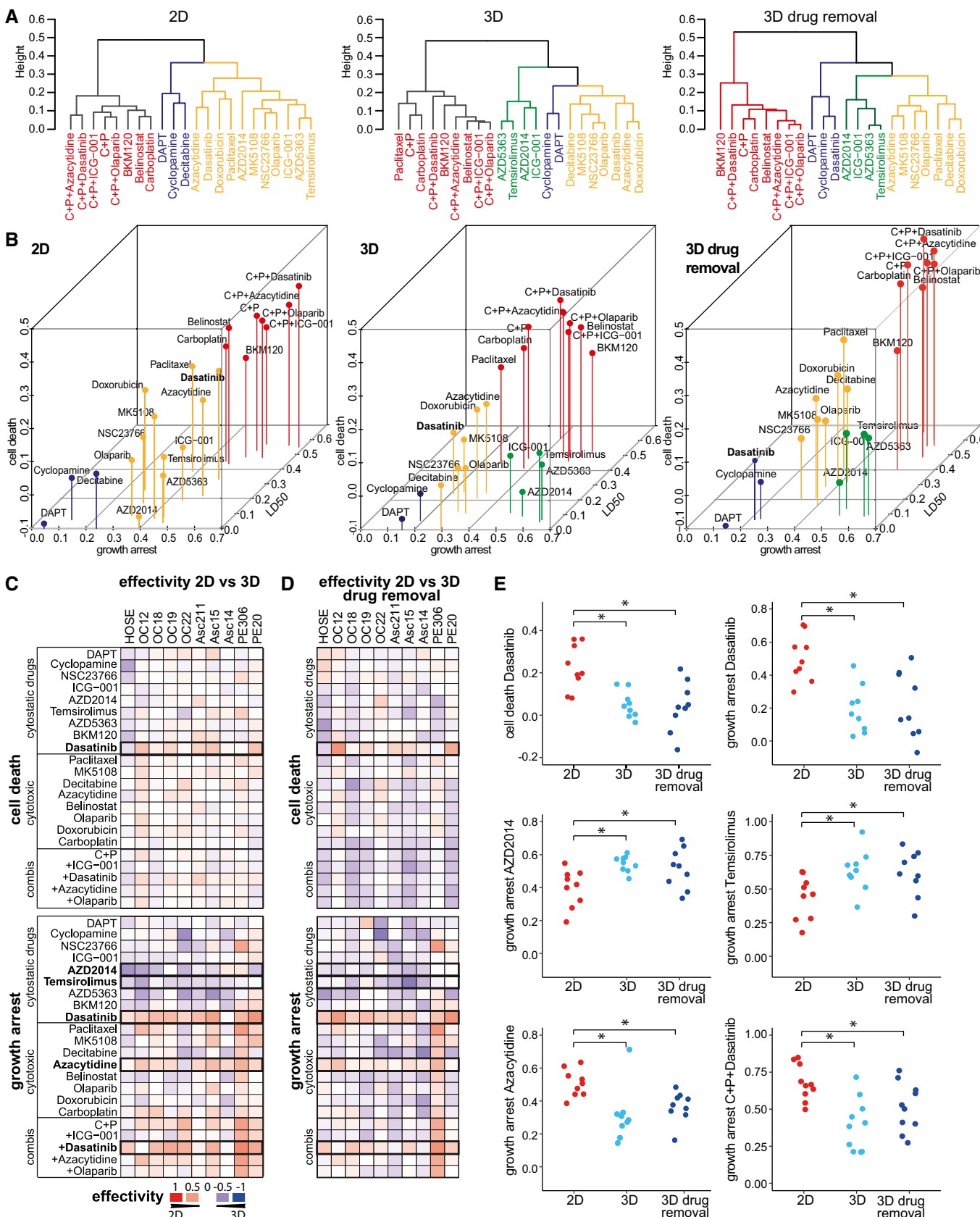

**Figure 3.**

◄

**Figure 3.  Culture type determines effectivity of targeted drugs like SRC inhibitor dasatinib and mTOR inhibitors AZD2014 and temsirolimus.**

A    Hierarchical clustering of drug effects determined in ovarian cancer *DeathPro* screens in 2D culture and 3D culture. Dendrograms derived from hierarchical clustering of drug effects averaged over all 10 patient cell lines (AUCd: cell death, AUCpi: growth arrest, LD50: lognorm LD50, scaled to 1 for minimum and 0 for maximum dose). Subclusters are differently coloured.

B    3D visualization of dendrograms shown in (A). Drug effects are averaged over all 10 patient cell lines. Drug groups derived from clustering are coloured similarly as in (A).

C    Differences of drug effects in patient cell lines measured with *DeathPro* assay in 2D or 3D culture after 72 h (dell death: AUCd, growth arrest: AUCpi). Blue heat map colour indicates a higher drug response in 3D culture and red colour a stronger effect in 2D culture. Black boxes mark drugs whose effects are significantly altered in cancer organoids compared to 2D cultured cells.

D    Differences of drug effects in patient cell lines cultured in 2D or 3D. Effects were measured with *DeathPro* assay directly after 72-h drug treatment in 2D or after 72-h treatment and 72-h drug removal in 3D culture. Black boxes mark drugs whose effects are significantly altered in cancer organoids compared to 2D cultured cells.

E    Drugs whose efficiency in inducing cell death or growth arrest is significantly changed when not applied in 2D but 3D culture. Effects of drugs marked in (C, D). Dasatinib, AZD2014 and temsirolimus target proliferation pathways (cytostatic drugs), azacytidine induced cell death (cytotoxic drug).

Data information: HOSE, HOSEpiC; C + P, carboplatin + paclitaxel: *P < 0.05 in two-sided Welch's *t*-test.

## Drug responses in patient organoids are more diverse and of lower therapeutic potential

Having compared drug effects generally and separately, we inspected differences and similarities of patient cell responses in 2D and 3D culture by hierarchical clustering. Interestingly, drug response profiles tended to cluster based on the patients as well as the culture format (Fig 4A), indicating that culture type can influence patient cell responses to the same extent as intrinsic tumour heterogeneity. Most 2D patient profiles clustered together homogenously, with the exception of OC12 and OC18 which showed comparable response profiles in 2D and 3D. In total, we found 2D drug profiles in four subclusters while 3D drug profiles occurred in eight subclusters, demonstrating once more that drug profiles appear more diverse in organoids. Normal HOSEpiC cells clustered separately from patient cells in 3D but showed a drug response similar to OC19 in 2D, suggesting that the culture format can conceal differences in genomic aberrations and gene expression.

In our screens, we included ovarian epithelial cells (HOSEpiC) to examine cytotoxicity induced in noncancerous cells. To normalize drug efficacy in PDCLs to HOSEpiC, we calculated the therapeutic index (TI) as the ratio of LD50 values from PDCL and HOSEpiC in both cultures (Fig 4B). TI patterns in cancer organoids were less favourable overall than in 2D culture (blue colour, Fig 4B). The most effective candidates, carboplatin and belinostat, had positive TIs and low toxicity, while low TIs for BKM120 reflected high toxicity in normal cells. To identify patient-specific treatment options, the drug with the highest TI can be selected for each individual, for example MK5108 for OC12. For some patient cells, for example Asc14, belinostat would be suggested from organoid testing but not from 2D cell testing where cell death was too low to determine an LD50. Even if drug-induced cytotoxicity differed only minimally between 2D and 3D cultured patient cells (Figs 2 and EV4), therapeutic potentials in OC patient organoids were altered distinctively. By taking into account the heterogeneity in drug responses, our *DeathPro* assay allows the systematic deduction of patient-specific treatment options across cell cultures and patient cell lines.

## Patient cells harbour numerous copy number alterations not linked to drug-induced cell death

To predict or functionally link drug sensitivities to genetic alterations, several studies have integrated drug sensitivity data from viability assays of patient cells or cell lines with genome sequencing data (Garnett *et al*, 2012; Van de Wetering *et al*, 2015; Schütte *et al*, 2017). Here, we performed whole-genome sequencing (WGS) to associate OC genotypes with drug sensitivity data. First, we confirmed that genetic alterations in our PDCL set matched those observed in tumours: we found multiple copy number alterations (CNA) in all PDCLs (Fig 5A), as previously reported for serous OC (Lambrechts *et al*, 2016). In a set of OC-relevant genes selected from literature (The Cancer Genome Atlas Research Network, 2011; Ciriello *et al*, 2013; Patch *et al*, 2015) and databases (Zhang *et al*, 2011; Forbes *et al*, 2015), few insertion/deletion polymorphisms (indels) or mutations were detected, except for *TP53*, which was mutated with a similar frequency as in the COSMIC cohort (Fig 5B). Likewise, genes frequently amplified (*MYC*, *PIK3CA* and *AURKA*) or lost (*RB1*, *PTEN*) in ovarian tumours (The Cancer Genome Atlas Research Network, 2011) were also commonly multiplied or lost in our set of patient cells (Fig 5B). Since drug sensitivity frequently correlates with alterations in the corresponding drug target (Garnett *et al*, 2012), we associated target genes commonly affected by CNAs with cell death (AUCd) induced by the respective inhibitor. Amplifications of *AURKA* and *PI3KCA* did not alter cytotoxicity induced by AURKA inhibitor MK5108 or PI3K inhibitor BKM120, respectively (Fig 5C and D). Moreover, loss of *BRCA1/2*, a putative marker for impaired DNA repair capacity (Abkevich *et al*, 2012), did not affect sensitivity towards the DNA damage-related drugs carboplatin and olaparib (Fig 5E and F).

## Homologous recombination deficiency scores correlate with drug effects in organoids

To incorporate the complex genomic aberrations in OC, we focused on the genome structure altered by DNA repair deficiencies. Loss of heterozygosity regions can be counted and added up to the homologous recombination deficiency (HRD) score (Fig 6A) which is linked to cellular HR repair capacity (Abkevich *et al*, 2012). HRD scores in our OC set varied between 3 and 22 (Fig 6B). We systematically associated HRD scores and OC drug responses in different culture systems and found 20 statistically significant correlations ($R^2 > 0.61$, false discovery rate < 0.1, Fig 6C). Remarkably, 90% (18/20) of these potentially relevant associations were observed with 3D culture-derived data. HRD scores correlated not only with cytotoxic responses to carboplatin and all its combinations (Fig 6C and D) but also with paclitaxel, azacytidine and decitabine

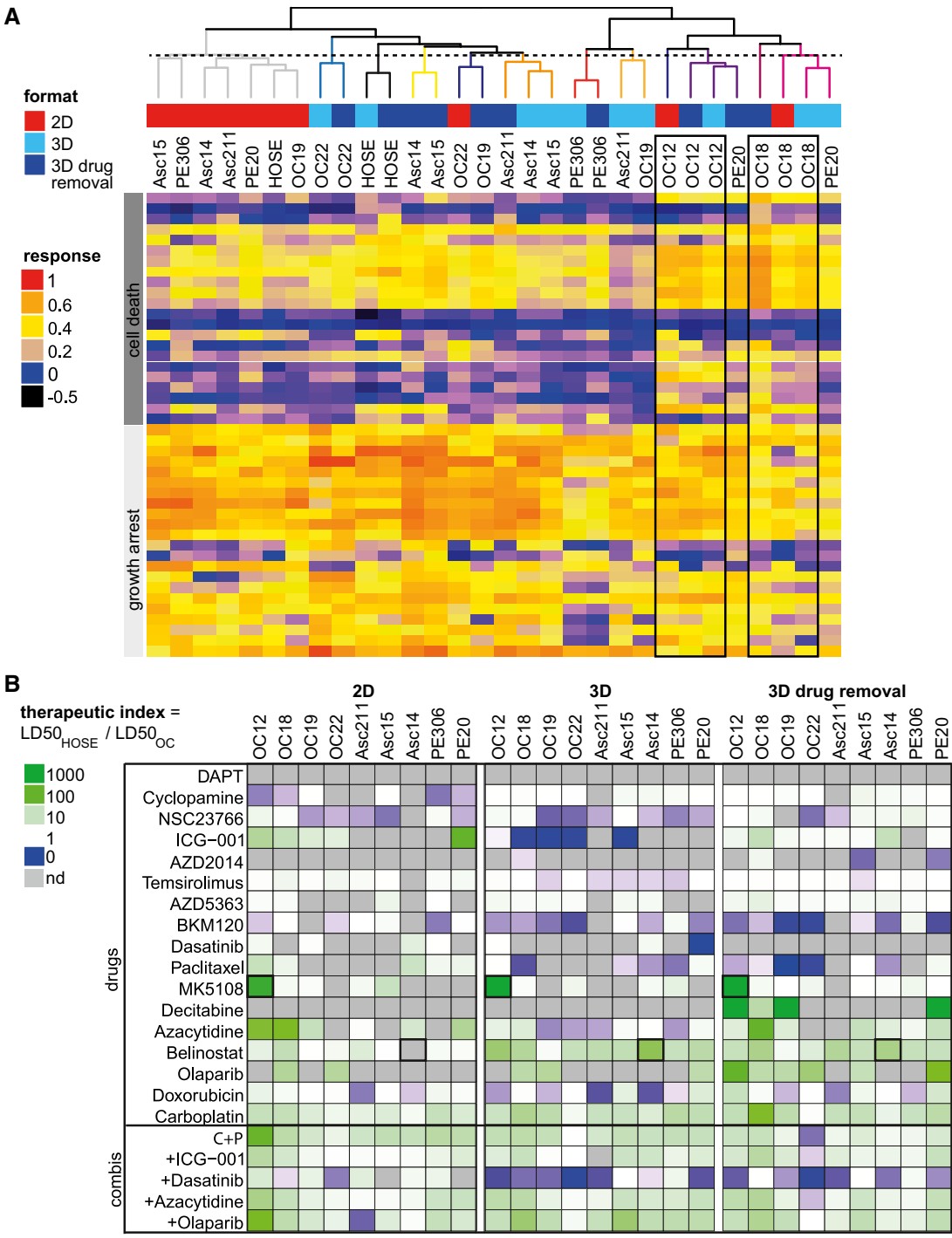

**Figure 4. Patient organoids respond more diverse to drugs and with lower therapeutic potential than 2D cultured patient cells.**

A   Hierarchical clustering of drug response profiles determined in ovarian cancer (OC) organoids or 2D cultured patient cells of the same origin. The dashed vertical line cuts the dendrogram arbitrarily at the height of the 2D subcluster (grey).

B   Therapeutic indices determined from LD50 values derived from OC drug screens in 2D or 3D culture. Green heat map colour indicates drug effectivity in cancer cells and low toxicity in normal cells, blue colour high toxicity in normal cells and low effectivity in cancer cells. nd, not determined, that is no fitting performed due to low response.

responses although these drugs do not directly affect DNA structure or repair (Fig 6E–G). Moreover, HRD scores were linked to growth arrest induced by temsirolimus (Fig 6H). Stratification based on

high ($\geq$ 10) or low (< 10) HRD scores divided OC cells into responders (OC12, OC18 and PE20) and non-responders to carboplatin, olaparib or azacytidine (Fig 6D and F, Appendix Fig S4A). The OC

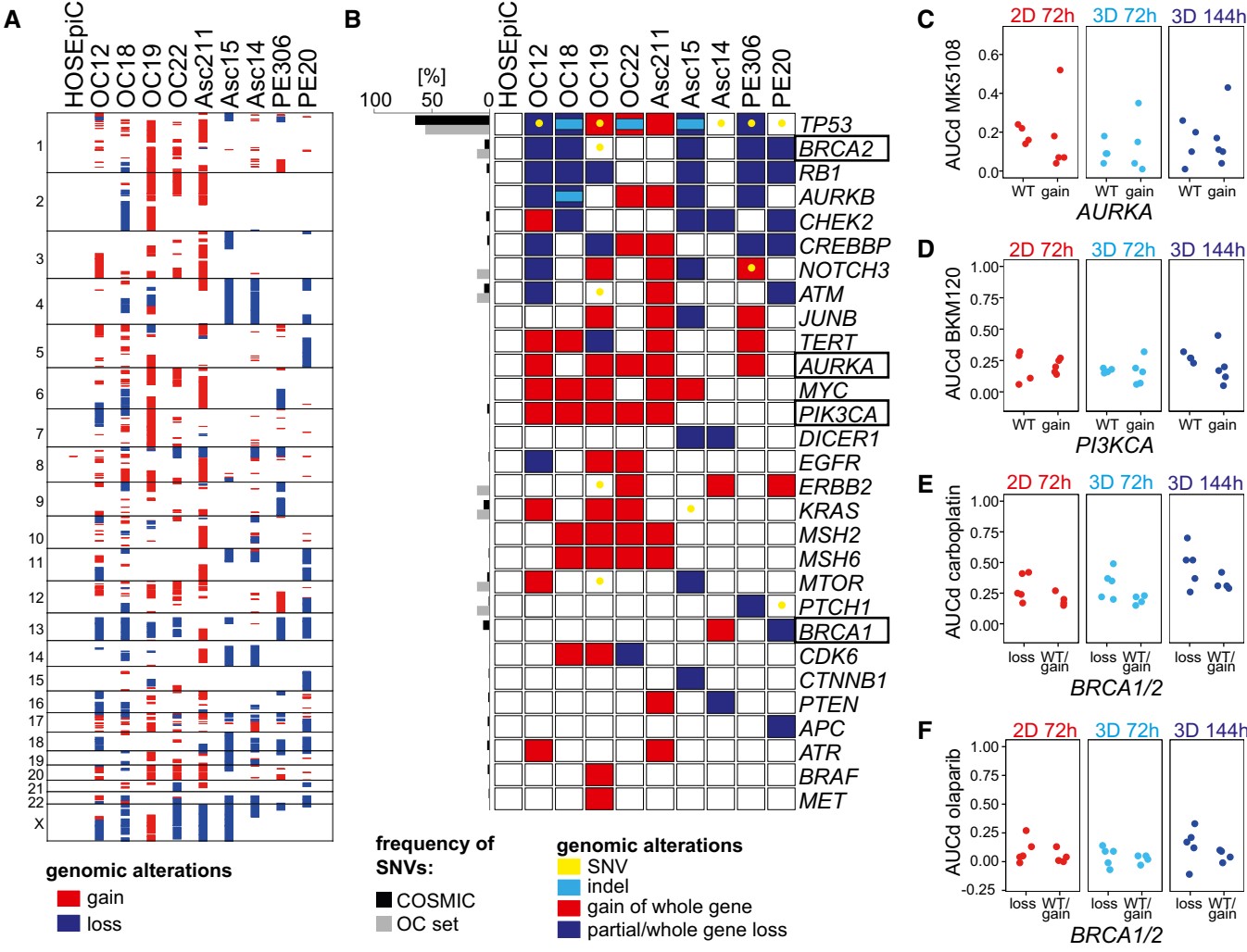

**Figure 5.  Ovarian cancer cells harbour numerous copy number alterations unlinked to drug-induced cell death.**

A    Copy number alterations (CNAs) in primary ovarian cancer (OC) cell lines used for *DeathPro* drug screening. CNAs were determined from whole-genome sequencing data. Losses are shown in blue, gains in red.

B    Panel of 29 OC-relevant genes depicting patient cell line-specific gains, losses including loss of heterozygosity, indels and somatic nucleotide variations (SNVs) in coding regions. Genes were selected from COSMIC and ICGC databases.

C–F  Association of copy number changes in drug target genes with drug sensitivities. (C) Cytotoxicity of Aurora kinase A inhibitor MK5108 in patients with or without (WT) *AURKA* amplification. (D) Cytotoxicity of PI3K inhibitor BKM120 in patients with or without (WT) *PI3KCA* gain. (E, F) Cytotoxicity induced by carboplatin (E) or olaparib (F) in patients with or without (WT) *BRCA1* or *BRCA2* loss. Two-sided Welch's *t*-test was performed.

responders grew faster than non-responders in organoid but not in 2D culture (Fig 2C and D, Appendix Fig S4B). Thus, high HRD scores co-occurred not only with high drug-induced cytotoxicity but also with fast growth in organoids. Altogether, the strong correlation of growth and HRD scores with drug response in cancer organoids supports our view that organoids are a better model to assess patient-specific drug response *in vitro*.

## Discussion

In this study, we systematically compared drug responses between 2D and organoid cultures of patient cells and their association with

genomic alterations. For this purpose, we developed *DeathPro*, an automated microscopy-based workflow that simultaneously discriminates cytotoxic and cytostatic drug effects over time. Previous microscopy-based drug assays in 3D cell cultures or organoids focused on morphological changes (Celli *et al*, 2014; Härmä *et al*, 2014), metabolic parameters (Walsh *et al*, 2016) or required specific instrumentation to resolve cell death and growth (Jung *et al*, 2016; Walsh *et al*, 2016). Tested in a small number of cell models and not in parallel in 2D cultures, the usability and scalability of these assays is limited (Celli *et al*, 2014; Jung *et al*, 2016; Walsh *et al*, 2016). We have demonstrated the versatility and robustness of *DeathPro* in drug screens of heterogeneous OC cells in monolayer and organoid culture, co-cultures with fibroblasts, PDX-derived cells

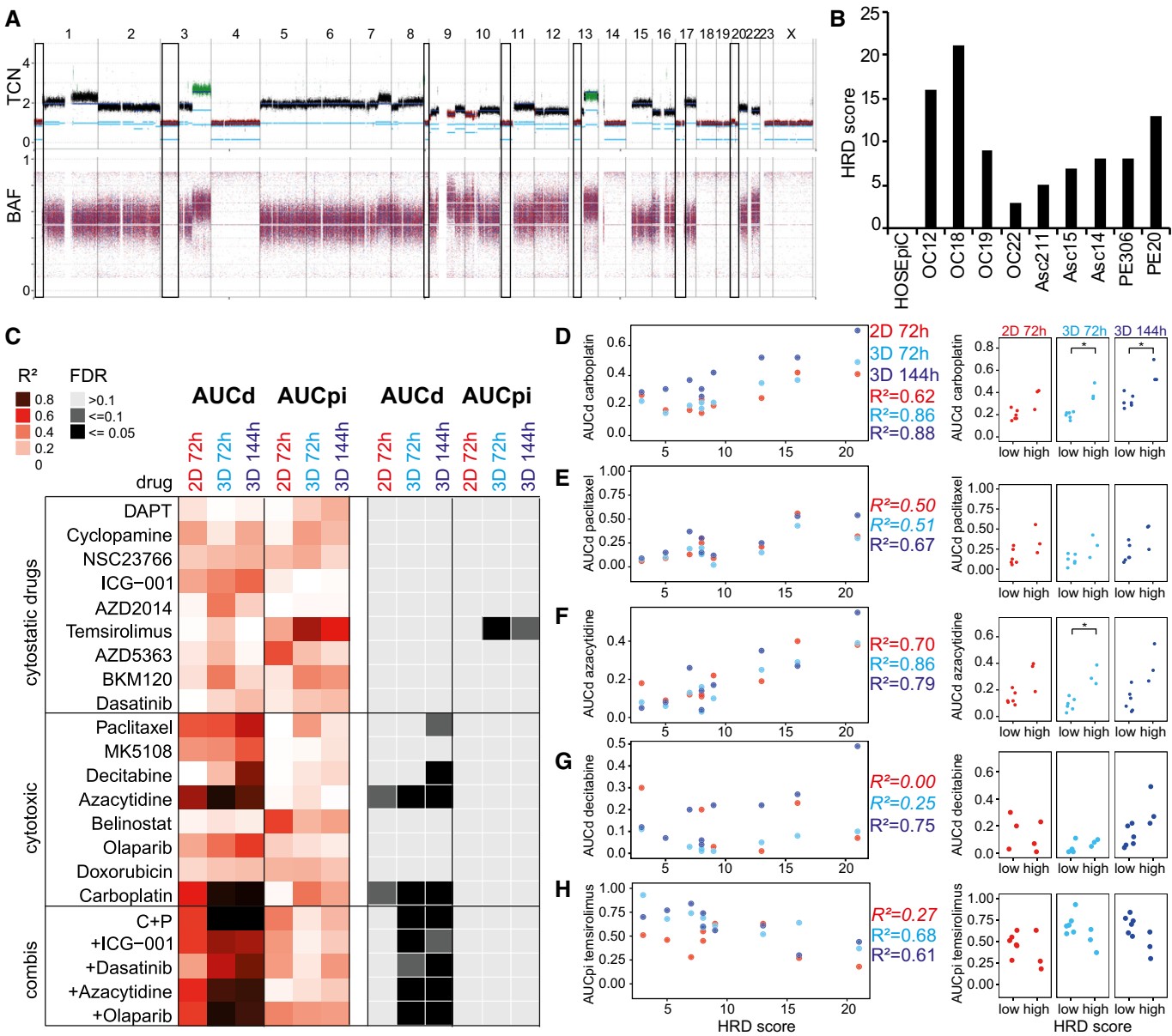

**Figure 6.  Homologous recombination deficiency scores correlate with drug-induced cell death in primary ovarian cancer cells.**

A    Visualization of homologous recombination deficiency (HRD) score determination by counting lost chromosome regions. Total copy number (TCN) and B-allele frequency (BAF) plots derived from Asc15 whole-genome sequencing data are shown. Black squares illustrate chromosome regions summarized as HRD score.

B    HRD scores of patient-derived ovarian cancer cell lines used for *DeathPro* drug screening.

C    Heat map of correlation coefficients ($R^2$) and estimated false discovery rates (FDR) determined from systematic association of drug responses (AUCd: cell death, AUCpi: growth arrest) with HRD scores. FDR was estimated by random sampling.

D–H    Drug-induced cell death (AUCd) or growth arrest (AUCpi) of all nine primary OC cell lines divided into two groups with low (< 10) or high (≥ 10) HRD score. Cytotoxicity induced by carboplatin (D), paclitaxel (E), azacytidine (F) and decitabine (G) correlates with HRD score. Growth arrest induced by temsirolimus (H) is reduced in HR deficient cells. $R^2$ values of not significant correlations are shown in italics.

Data information: *P < 0.05 in two-sided Welch's t-test.

as well as in lung cancer cells as a second cancer entity. Unlike most image-based viability assays, which detect viable cells by cytoplasmic staining with Calcein AM (Celli *et al*, 2014; Trumpi *et al*, 2015), *DeathPro* directly compares the area of nuclei of dead and live cells and generates drug efficacy measures over time independent of cellular morphology and cytoplasmic stains. Counting dead

and live cells as an alternative to area measurements would require detailed, time-consuming imaging of organoids unfeasible in high-throughput drug screens. To the other end, subtle changes in nuclear size due to mitosis defects and apoptosis might be neglected. Even though we here presented drug screens based on Hoechst for staining live cells, the *DeathPro* image analysis

workflow provided can be readily adapted to other nuclear stains or markers.

Resolving drug effects in OC patient cells, we found that drug-induced cell death was similar in both culture types whereas growth arrest varied. Accordingly, the efficacy of cytostatic drugs like dasatinib, temsirolimus or AZD2014 was culture type-dependent. Since most newly developed drugs are cytostatic (Steeg, 2016), our results highlight the importance of choosing the right model system to evaluate drug efficacy, for example in preclinical studies. In particular, our results reveal diverse drug responses in organoids and suggest that specific drug response phenotypes are visible in organoids but not in monolayer culture. We observed less cell death in 3D compared to 2D cultures after 72 h but higher death after drug removal (114 h), which may lead to an underestimation of drug effects in 3D after a 72-h standard treatment interval. The previously reported findings that standard cell lines in 3D culture are more chemoresistant than in 2D culture (Lee et al, 2013; Edmondson et al, 2014) may therefore in part reflect altered cell death kinetics, which should be accounted for in future screens. Interestingly, the observed drug effects in OC patient cells mirrored findings from clinical trials. The combination of carboplatin and paclitaxel did not perform better than carboplatin alone, consistent with the ICON 3 trial (Parmar et al, 2002). Paclitaxel killed only two of nine patient cancer organoids, similar to taxol monotherapy studies in metastatic or refractory OC that reported 20% responders (Einzig et al, 1992; Trimble et al, 1993). Dasatinib, which failed at clinical phase II for recurrent OC and primary peritoneal carcinoma (Schilder et al, 2012), was effective in 2D but ineffective in 3D culture in our screen. From the drugs included in our panel, no candidate surpassed the first-line therapeutic carboplatin with regard to (i) efficacy in the whole patient set, and (ii) limited toxicity in normal epithelial cells. Still, initial cytotoxicity profiles determined with DeathPro readily suggested patient-specific alternatives to carboplatin, such as Aurora kinase A inhibitor MK5108 for chemosensitive patient OC12 or belinostat for chemoresistant patient Asc14.

To the best of our knowledge, we provide the first detailed comparison of drug effects in primary cells cultured in the absence or presence of an extracellular matrix. By seeding patient-derived cells onto Matrigel, we generated OC organoids that, like other cancer organoids, lack stromal and immune cells as well as functional vasculature (Van de Wetering et al, 2015; Pauli et al, 2017; Schütte et al, 2017). As drug effects changed upon addition of fibroblasts to 2D cultured patient cells, immune and stromal cells might also profoundly alter organoid drug responses. In the future, the OC organoid model could be enhanced by including stromal and metastases relevant cells such as mesothelial cells (Yeung et al, 2015).

By associating resolved drug responses in OC patient cells with HRD scores from WGS data, we found a set of relevant correlations which would not be detected with proliferation-based assays in 2D. As expected based on a study that linked higher platinum sensitivities to HR deficiency (Telli et al, 2016), carboplatin-induced cytotoxicity correlated with HRD scores in patients with OC. Moreover, we detected correlations of DNA demethylating drug effects with HRD scores, suggesting a link between deficient DNA homologous recombination repair and DNA demethylation. While only decitabine sensitivity has been linked to KRAS status so far (Stewart et al, 2015), there is increasing evidence that both azacytidine and decitabine induce reactive oxygen species which cause DNA damage and

finally apoptosis in cancer cells (Gao et al, 2008; Shin et al, 2012; Fandy et al, 2014). For all drugs whose effects correlated with HRD score, we observed a stronger correlation in OC cancer organoids than in monolayer culture. Interestingly, cell growth, HRD score and drug-induced cytotoxicity were linked in organoids but not in 2D cell culture. Similar to drug efficacy, this suggests that genotype–drug sensitivity correlations are more pronounced in 3D cultures, which is particularly important since comprehensive studies so far have focused on 2D culture data (Barretina et al, 2012; Garnett et al, 2012).

Taken together, we developed and provide the DeathPro assay as a tool for refined drug screening and for deciphering genotype–drug sensitivity associations, and found that culture type was a key determinant of the efficacy of cytostatic drugs. In our hands, drug sensitivity was not generally decreased in organoids as previous studies suggested; instead, drug responses were more diverse and correlated better with genomic alterations in 3D compared to 2D culture. Overall, these results could provide a rationale to select the appropriate culture format for drug sensitivity assays in basic and future translational research.

## Materials and Methods

### Patient-derived cell lines and organoids

Tumour material from serous ovarian cancer patients was collected at the Departments of Gynaecology and Obstetrics, at the University Medical Centres Mannheim and Heidelberg. The study was approved by the ethical committees of the Universities of Mannheim and Heidelberg (case number 2011-380N-MA and S-008/2009) and conducted in accordance with the Helsinki Declaration; written informed consent was obtained from all patients. Primary serous ovarian carcinoma cell lines except for Asc211 and PE306 were established by transplantation of primary tumour specimen or tumour cells as previously described (Aloia et al, 2015; Noll et al, 2016). In detail, xenografts were established by first cutting primary serous adenocarcinomas into pieces < 2 $mm^3$ and then transplanting them subcutaneously into NOD.Cg-Prkdc[scid] Il2rg[tm1Wjl] NSG mice. Ascites or pleural effusion samples were spun down, remaining erythrocytes were removed using ACK buffer (Lonza), and the resulting cell suspension was then filtered through a 40 μm mesh (Greiner Bio-One). For initiation of xenografts, at least $1 \times 10^6$ cells were injected intraperitoneally into NOD.Cg-Prkdc[scid] Il2rg[tm1Wjl] NSG mice. Mice were monitored for several months until tumour engraftment was detected. For establishment of OC PDCLs except Asc211 and PE306, engrafted tumours were taken out, cut into pieces < 1 $mm^3$ and then enzymatically disaggregated into a single cell suspension with 1 μg/ml collagenase IV (Sigma) and DNase (Sigma) or with the human tumour dissociation kit (Miltenyi Biotec) for 2 h at 37°C on a MACSMix rotator (Miltenyi Biotec) with occasional vortexing. Remaining erythrocytes were removed using ACK buffer. The resulting suspension was then filtered through a 40 μm mesh.

Cell lines were initiated by plating single cell suspensions ($0.5 – 1 \times 10^5$ cells) in T25 PRIMARIA flasks in a defined serum-free culture medium as described in (Noll et al, 2016) with the addition of 36 ng/ml hydrocortisone (Sigma), 5 μg/ml insulin (Life

Technologies) and 0.5 ng/ml beta-estradiol (Sigma), referred to as CSC medium. For initial cell growth, CSC medium was supplemented with 50 µg/ml gentamicin (Life Technologies), 0.5 µg/ml Fungizone (Life Technologies) and 10 µM ROCK inhibitor Y27632 (Selleckchem). Adherent monolayer cultures were maintained and incubated at 37°C and 5% $CO_2$, and all subsequent passages were propagated without antibiotics/ROCK inhibitors. Contaminating fibroblasts were removed by sequential differential enzymatic digestion with StemPro Accutase (ThermoFisher). Asc211 and PE306 cell lines were established directly from patient material. Cell suspensions were prepared as described above and taken directly into 2D culture. Tumorigenicity of PDCLs was verified by injecting $1 \times 10^6$ cells intraperitoneally into NOD.Cg-*Prkdc*[scid] *Il2rg*[tm1Wjl] NSG mice and assessment of tumour growth. HOSEpiC cells were obtained from ScienCell Research. PDCLs were checked for cross-contamination with standard OC cell lines and tested for mycoplasma contamination using the commercial Multiplex Cell Line Authentication and Mycoplasma Test Services (Multiplexion, Heidelberg, Germany). All OC cells were used at passages below 20 (PDCLs) or 6 (HOSEpiC).

To generate organoids, PDCLs and HOSEpiC were seeded onto growth factor reduced, phenol red-free Matrigel (Corning, > 9 mg/ml protein) using CSC medium supplemented with 2% (v/v) Matrigel to a density of 5,000–12,500 cells/cm². Organoids were grown for up to 10 days, and medium was renewed every 3–4 days to CSC medium without Matrigel.

To test *DeathPro* in cells directly derived from xenografts, 10,000 OC12 cells were intraperitoneally injected into NOD.Cg-*Prkdc*[scid] *Il2rg*[tm1Wjl] NSG mice to form ascites. Cell clusters washed out from mice ascites were subjected to erythrocyte lysis and seeded onto Matrigel in CSC medium supplemented with antibiotics.

The cell lines LN2106 and T2427 were generated from human squamous cell lung carcinomas as described previously (Gottschling *et al*, 2012). Their use for research was approved by the ethical committee of the University of Heidelberg (S-270/2001). LN2106 and T2427 cells were cultivated in DMEM/Ham's F-12 (Thermo-Fisher) with 10% foetal calf serum (ThermoFisher) for not more than 20 passages. Lung cancer organoids were generated similar to OC organoids.

## DeathPro microscopy-based drug screens

Drugs were dissolved in DMSO, water, PBS or ethanol and stored as single-use aliquots at −80°C (Table EV2). Drug dilution series (1:3) were prepared using the respective culture medium. For drug combinations, two or three drugs were combined by using similar concentrations as for single drug testing. Drug concentrations, treatment intervals and endpoints were chosen according to published studies or determined in pilot experiments. Drug screening was performed in 96-well Angiogenesis µ-Plates from ibidi. For 2D culture screens, 5,000 OC cells per well were seeded in 70 µl CSC medium directly onto the plate. For organoid screens, 2,500 cells were seeded in CSC medium containing 2% Matrigel onto 10 µl solidified Matrigel. Drugs were added to CSC medium containing 1 µg/ml Hoechst (Invitrogen) and 1 µg/ml PI (Sigma) 1 day (2D) or 4 days (3D) after cell seeding. After 72 h, organoids were washed twice with PBS and drug-containing medium was substituted by drug-free medium. Likewise, lung cancer cells were seeded onto Matrigel, treated with drugs in Hoechst- and PI-containing medium

from day 4 to day 7 after seeding and incubated for another 72 h until day 10 in drug-free medium.

For the 2D OC co-cultures, 1,000 primary human ovary fibroblasts or IMR90 lung fibroblasts stained with 1 µM CellTracker Green (ThermoFisher) for 40 min were seeded together with 2,000 OC12 cells or 4,000 cells from OC15, OC20 or PE306 in 70 µl CSC medium per well. Drug treatment in CSC medium started at day 1 after seeding and lasted for 72 h.

In each drug assay, cells were exposed maximally to 1% DMSO or 1% ethanol in the highest drug concentrations and corresponding controls were included in the assay.

Cells were imaged at similar positions at 0, 72 and 144 h (only 3D) after start of drug treatment using a Zeiss LSM780 confocal microscope, 10× objective (EC Plan-Neofluar 10×/0.30 M27) and 405 and 561 nm diode lasers in simultaneous mode. Imaging was performed in an incubation chamber at 37°C, 5% $CO_2$ and 50-60% humidity using the Visual Basic for Applications macro "Autofocus-Screen" (Conrad *et al*, 2011). All image data were used and analysed. To assess reproducibility, the drug screens in OC cells and organoids were performed twice independently with different cell passage numbers and different drug plate layouts. Lung cancer organoids were screened once. Biological variability in all tested conditions was assessed by imaging two positions per well, and no other technical replicates were included.

## Image processing and drug response analysis

Image stacks were processed to maximum intensity projections (MIPs) with a custom-built macro "MIP_export.ijm" in Fiji 2.0.0-rc-19/1.49 m (Schindelin *et al*, 2012). MIPs were uploaded and processed in our "*DeathPro*" workflow in KNIME 3.1 (Konstanz Information Miner, Berthold *et al*, 2008). In short, images were annotated with drugs and concentrations used, smoothed by median filtering and signals extracted by local mean thresholding. Sums (areas) of binary images were calculated and used to calculate cell death and growth using R version 3.3.2 (R Core Team, 2016; see detailed Information). For the calculation, summarizing, clustering and plotting of values, the packages drc (Ritz *et al*, 2015), stringr, Complex-Heatmap (Gu *et al*, 2016), ggplot2 (Wickham, 2009), reshape (Wickham, 2007) and RColorBrewer (Neuwirth, 2014) were used.

Hierarchical clustering with Euclidian distance and complete linkage was used to compare PDCL-specific drug response profiles consisting of cell death (AUCd) and growth arrest (AUCpi) values measured over all drugs tested. Average linkage was used for drug response parameters averaged over all PDCLs.

The "*DeathPro*" workflow including R-scripts, Fiji macro, installation manual, example data and results can be found in Code EV1.

## Image processing in the DeathPro KNIME workflow

The KNIME workflow has been optimized for images acquired as follows: to minimize laser intensities and phototoxicity during drug screen imaging, a large pinhole (172 µm 2D, 200 µm 3D) was used. Image stacks with 15–20 slices (3D culture) or seven slices (2D culture), 50 µm slice distance and 512 × 512 pixels with a resolution of 2.767 µm/pixel were acquired simultaneously with 405 and 561 nm lasers. This coarse confocal imaging strategy allows scanning of a 96-well plate with two positions/well within 40 min (2D)

or 70 min (3D culture). The 405 nm laser intensities were adjusted for each PDCL so that the signal-to-noise ratio for Hoechst intensities in live cells to background was greater than 2. Intensity of the 561 nm laser was chosen in a way that PI intensities were not saturated. At each time point analysed, the plate was calibrated and images were taken at the same position in the wells.

In the KNIME workflow, the MIPs of one plate are loaded and annotated using a "plate layout" csv-file chosen by the user. All MIPs were smoothed using median filtering with a radius of 0.5 pixels. To cope with varying cell and organoid morphologies, image-derived parameters were used for thresholding. For each plate, images from untreated control were filtered. From these images, the radius of the largest cancer organoids or cells was determined using mean local thresholding and the initial radius defined at beginning. For the Hoechst images, two thresholds were applied: a first mean local threshold with the large radius determined from the controls to detect big structures followed by a second local threshold with small radius to get small structures of low contrast (live cells). For the PI channel, only one threshold was used since PI stains only dead cells, whereas Hoechst intensities are high in dead and low in live cells. Images with overall changes in Hoechst intensities caused by high drug concentrations were filtered out and subjected to local thresholding with adapted sensitivity parameters. After thresholding, the binary images of Hoechst and PI were combined (H + PI) to calculate the area of all cells. Small objects (artefacts) were filtered out from binary images. The area (sum) of dead cells (PI) or all cells (H + PI) was determined and exported in csv-files. R-scripts (see below) were used to finally calculate all screen parameters.

In the 2D co-culture screens, signals from green fluorescent fibroblasts were acquired subsequently to Hoechst and PI signals with a 488 nm laser. For segmentation, a mean local threshold (radius = 35 pixel, $c = -2$) was used and binary images were used to filter out signals from fibroblast nuclei. Thus, cell death and growth arrest were determined exclusively for OC cells.

## Cell death and growth inhibition analysis in R

The csv-files generated by KNIME that contained the area values were used to calculated LD50 and AUCd as follows. Drug response curve fitting to determine the LD50 was only performed if there was a significant difference between cell death in drug-treated and untreated samples. Therefore, an analysis of variance (ANOVA) was performed and LD50 was only calculated for drugs with *P*-values < 0.0005. Drug response curve fitting and LD50 calculation were performed using the LL2.4 function of the drc package with Hill Slope > 0 and 1 as maximum value for death ratio. AUCd values were calculated with the following reference: median of the death ratio in controls. AUC values were normalized as follows: AUC > 0 to 1-median and AUC < 0 to the median. In late drug response time points like 144 h, cells are often dead for several days, disintegrated and thus can be hardly stained with PI. Cells weakly stained with PI might not be detected and mislead the results. However, dead cells cannot move away and thus the area of dead cells cannot decrease from an earlier to a later time point. This rationale was applied for the correction of 144 h values: PI area values determined after 72 and 144 h at the same position were compared and set to the 72-h value in case dead cells were lost (144 h value < 72 h value).

Cell growth was calculated by dividing the area of all cells (H + PI) of the later time point (72 or 144 h) by the area of all cells (H + PI) at drug test start (0 h). The variation in growth due to image processing and confounding factors like migration or dispersion of organoids upon cell death necessitates filtering and adjustment of growth values. Otherwise artefacts in one of 16 images in one drug dilution series can completely distort the AUCpi of this drug. "Loss of dead cells" as described above was similarly corrected: growth after 144 h was set to growth after 72 h, in case growth 144 h < growth 72 h. Growth values smaller 1 were set to 1 to correct for confounding factors like cells migrating out of the position repeatedly imaged. The maximum growth value to filter out artefacts was set to 8 as this value refers to three cell doublings within 3 days. Furthermore, the median growth in controls (mgc) was determined for untreated, DMSO and ethanol-treated samples and used to limit growth under drug conditions to the twofold mgc. AUCpi values were calculated with mgc as reference. The described R code is included in the "*DeathPro*" KNIME workflow in Code EV1.

## Whole-genome sequencing and analysis

Genomic DNA from $1 \times 10^6$ primary cells was extracted using the DNeasy Blood & Tissue Kit (Qiagen), prepared with the TruSeq PCR free library kit (Illumina) and sequenced on a HiSeq X Ten (Illumina). Sequences were mapped to the human reference genome (build hg19, version hs37d5; McVean *et al*, 2012) using bwa-mem 0.7.8-r455 (Li, 2013). The OC22 sample contained ~30% mouse gDNA due to irremovable, immortalized mouse fibroblasts potentially derived from the mouse xenograft and thus had to be aligned to the hs37d5-mm10 hybrid reference sequence. Only reads mapped against hs37d5 were used for further analysis. WGS data of all other PDCLs and HOSEpiC contained only human DNA sequences. Duplicates were marked with Picard 1.125 (https://broadinstitute.github.io/picard/). Somatic nucleotide variations and indels were called without matched control using our in-house workflow (Jones *et al*, 2012), filtered (ENCODE Project *et al*, 2012) and annotated with Annovar (Wang *et al*, 2010; see below for detailed description). Copy number variations and loss of heterozygosity regions were determined by dedicated workflows, and gains and losses were classified based on estimated ploidies (see detailed information below). Genes listed in oncoprints were selected from top 100 altered genes in OC in COSMIC (Forbes *et al*, 2015) and ICGC (Zhang *et al*, 2011). Homologous recombination deficiency scores were determined as previously described (Abkevich *et al*, 2012) with the following changes: copy neutral loss of heterozygosity regions was excluded, chromosome 17 included and the region length was decreased from 15 to 10 Mb.

## SNV/indel calling from whole-genome sequence data

We called SNVs and indels from all 10 samples without matched control using our bioinformatic workflow (Jones *et al*, 2012, 2013). SNVs were identified from tumour samples by using samtools/bcftools version 0.1.19 (Li *et al*, 2009), and indels were determined by Platypus version 0.8.1 (Rimmer *et al*, 2014). For later selection of SNVs, a confidence score of 10 was set and further deducted if the SNV was part of repeats or listed in DUKE excluded regions, DAC blacklisted regions, self-chain regions or segmental duplication records as introduced in the ENCODE project (ENCODE Project *et al*,

2012). For indels, filters from Platypus were used to calculate a confidence score ranging from 0 to 10. SNVs and indels were excluded from the analysis if the confidence score was less than 8, sequencing depth was too high (> 150 reads) or too low (< 6 reads), or the reads were not properly mapped (according to the bwa-mem aligner). Moreover, we removed very common SNVs/indels that are potential polymorphisms as follows: first, mutations that could be found in dbSNP version 147 (Sherry *et al*, 2001) with "COMMON = 1" tag were removed, but rescued if they had a corresponding OMIM record in dbSNP. Then, we additionally removed mutations found in ExAC version 0.3.1 (> 0.1%; Lek *et al*, 2016), EVS (> 1%; Exome Variant Server, NHLBI Exome Sequencing Project (ESP)) and our control dataset (> 2%, among 280 controls). We used functional annotations from Annovar (release Feb 2016 Wang *et al*, 2010) to select only SNVs and indels found in coding regions for the oncoprint.

### Identifying structural variations and copy number variations

Without matched controls for the 10 ovarian cancer samples, structural variations (SV) and CNVs were determined by two bioinformatic workflows, named SOPHIA and ACEseq, respectively (manuscripts for both workflows in preparation).

Briefly, SOPHIA uses the "supplementary alignment" feature of the bwa-mem aligner, providing candidate chimeric alignments of reads which cannot be represented by a linear alignment because parts map to different locations in the genome ("split reads") which is an error-prone indicator of structural variations (SV). SOPHIA uses a decision tree to designate high-quality reads and low-quality reads that fall on poorly mappable regions or appear due to low-quality base calls. From the remaining high-quality reads, SOPHIA filters the results provided by the supplementary alignments generated by the aligner using control (blood) sequencing data from a large background population database of 1,740 patients across different diseases (published TCGA cohorts and published/unpublished DKFZ cohorts) and sequencing technologies (100 bp read length Illumina HiSeq 2000/2500 and 151 bp read length Illumina HiSeq X Ten) aligned using the same alignment settings and workflow. An SV is discarded if (i) the ratio of low-quality reads supporting one of the breakpoints exceeds 0.5, (ii) if the SV is detected only on one breakpoint (with the second either unmappable or undetected) and the exact same breakpoint was detected in more than three cases in the 1,740 patient population background model, (iii) the SV is detected by two breakpoints and one of them was exactly detected in more than 3% of the 1,740 patient population background model, (iv) both of the detected breakpoints had less than 10% allele frequency.

ACEseq (allele-specific copy number estimation from sequencing) uses the tumour coverage as well as the B-allele frequency (BAF) to determine copy numbers. In addition, tumour cell content and ploidy are estimated. During pre-processing of the data, allele frequencies were obtained for all single nucleotide polymorphism (SNP) positions recorded in dbSNP version 135 (Sherry *et al*, 2001). Positions with BAF between 0.1 and 0.9 in the tumour are assumed to be heterozygous in the germline. To improve sensitivity with regard to imbalanced and balanced regions, heterozygous and homozygous alternative allele SNP positions were phased with impute2 (Howie *et al*, 2009; McVean *et al*, 2012). Additionally, the coverage for 10-kb windows with sufficient mapping quality and read density in a control was recorded for the tumour and

subsequently corrected for GC-content and replication timing to remove coverage fluctuations caused by these biases. The genome was segmented using the PSCBS package in R (Van Den Meersche *et al*, 2009; Olshen *et al*, 2011) while incorporating SV breakpoints defined by SOPHIA. Segments were clustered according to their coverage ratio and BAF value using k-means clustering. Neighbouring segments that fell into the same cluster were joined. Small segments were attached to the more similar neighbour. Finally, tumour cell content and ploidy of the samples were estimated by fitting different tumour cell content and ploidy combinations to the data. Segments with balanced BAF were fitted to even-numbered copy number states whereas unbalanced segments were allowed to fit to uneven numbers as well. Lastly, estimated tumour cell content and ploidy values were used to compute the total and allele-specific copy number for each segment. Based on the ploidy values from ACEseq, we set thresholds for copy number gain/loss for $2n$ PDCLs to > 2.7, < 1.3 and for $3n{-}4n$ > 4.7, < 2.3. In addition, loss of heterozygosity was determined when total copy number of one allele was lower than 0.5. For OC22, due to the mouse genome contamination, we set the threshold for LOH to 1.5. For further analysis, we selected fully inclusive genes in the segments from the gain/loss segments, based on the BioMart (Kasprzyk, 2011) dataset from Ensembl release 85 (Yates *et al*, 2016).

### Statistical analysis

Independent replicates refer to independent cell samples seeded, treated and imaged on different days. Differences between effects of drug combinations and single drugs were tested for statistical significance using a paired Student's *t*-test. Differences between responses of different groups to one drug were assessed with a two-sided Welch's *t*-test. *P*-values < 0.05 were considered statistically significant and indicated with asterisks. Pearson's correlation coefficient (Rp) was used to describe the strength of correlation between biological replicates. Coefficient of determination ($R^2$) was used to denote strength of linear relationships between area under curve values and HRD scores. False discovery rate for $R^2$ was determined by random sampling.

### Data availability

Sequence data have been deposited at the European Genome-phenome Archive (http://www.ebi.ac.uk/ega/), which is hosted by the EBI, under Accession Number EGAS00001002239.

**Expanded View** for this article is available online.

### Acknowledgements

We thank A. Kopp-Schneider (Division of Biostatistics, DKFZ) for counselling statistical analysis, C. Dietz (Konstanz University), M. Waschow and L. Mayer (Theoretical Bioinformatics, DKFZ) for support with KNIME, B. Burwinkel (Molecular Epidemiology, DKFZ) for provision of several ovarian cancer samples, D. Hübschmann, G. Warsow (Theoretical Bioinformatics, DKFZ) for assistance in WGS data analysis, C. Klein and V. Vogel for immunohistochemistry and T. Krieger, M. Waschow and D. Niopek (Theoretical Bioinformatics, DKFZ) for critically revising the manuscript. We thank the High Throughput Sequencing unit and the Microarray unit (Genomics & Proteomics Core Facility, DKFZ) for providing whole-genome sequencing and microarray services. Tissue

samples were provided by the tissue bank of the National Center for Tumour Diseases (NCT, Heidelberg, Germany) in accordance with the regulations of the tissue bank and the approval of the ethics committee of Heidelberg University. The authors would like to thank the Exome Aggregation Consortium and the groups that provided exome variant data for comparison. A full list of contributing groups can be found at http://exac.broadinstitute.org/about. JJ is a recipient of the Annemarie Poustka fellowship of the German Cancer Research Center. This study was supported by the BMBF-funded Heidelberg Center for Human Bioinformatics (HD-HuB) within the German Network for Bioinformatics Infrastructure (de.NBI) (#031A537A, #031A537C), the Helmholtz International Graduate School for Cancer Research, DKFZ NCT3.0 Precision Oncology Program PANC-STRAT (NCT3.0_2015.16 PDAC POP), the BMBF-funded e:Med program for systems medicine (PANC-STRAT, 01ZX1305) and in part by the Dietmar Hopp Foundation, and the Swiss Bridge Foundation and iMed Program (Helmholtz Association) (to AT). DKFZ-HIPO provided technical support and funding through Grant No. HIPO 059.

## Author contributions

JJ, RE and CC conceived the study. JJ, CC, MSc, MSp, AT and RE designed experiments. JJ performed the drug screens and most of the experiments and analysed image data. SW, FMZ and KJ generated and characterized the PDCLs. JP, KK and UHT analysed WGS data; JP and JJ analysed drug–genome associations. XJ performed drug response curve fitting and provided statistical advice. MASc, MM, SS and MSu contributed crucial samples and clinical expertise. JJ and CC wrote the manuscript. All authors revised and approved the manuscript.

## Conflict of interest

The authors declare that they have no conflict of interest.

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
