## [Review Process File · Molecular Systems Biology]

Screening drug effects in patient-derived cancer cells links organoid responses to genome alterations

Julia Jabs, Franziska M. Zickgraf, Jeongbin Park, Steve Wagner, Xiaoqi Jiang, Katharina Jechow, Kortine Kleinheinz, Umut H. Toprak, Marc A. Schneider, Michael Meister, Saskia Spaich, Marc Sütterlin, Matthias Schlesner, Andreas Trumpp, Martin Sprick, Roland Eils & Christian Conrad

Corresponding authors: Roland Eils and Christian Conrad, German Cancer Research Center (DKFZ) and University of Heidelberg

Review timeline:

Submission date:	18 April 2017
Editorial Decision:	02 June 2017
Revision received:	20 September 2017
Editorial Decision:	24 October 2017
Revision received:	26 October 2017
Accepted:	27 October 2017

Editor: Maria Polychronidou

Transaction Report:

1st Editorial Decision

02 June 2017

Thank you again for submitting your work to Molecular Systems Biology. We have now heard back from the three referees who agreed to evaluate your study. As you will see below, the reviewers appreciate that the presented approach seems interesting and potentially useful for future analyses. However, they raise a series of concerns, which should be carefully addressed in a revision of the manuscript.

The reviewers' recommendations are clear so there is no need to repeat all the points listed below. As reviewer #2 points out, the addition of further experiments demonstrating that the method works with organoids derived directly from patient tumors and patient derived xenografts and the inclusion of analyses of co-cultures would significantly enhance the impact of the study. Moreover the reviewers mention several technical issues that need to be addressed and refer to the need to describe the experimental processes and methodology in better detail.

 REVIEWER COMMENTS

Reviewer #1:

Neugebauer et al. develop and implement an imaging-based method for evaluating drug effects on 2D and 3D spheroid cancer cell cultures systematically. They use this approach to show differences in drug effects on cells under different conditions and to relate these different drug actions to the genetic state of various patient derived cultures. The general field of in vitro drug evaluation and application to personalized medicine are of great interest to the medical field as well as basic scientists. Although the concept of differential drug effects in 2D vs. 3D culture is hardly novel, I am impressed by the analytical rigor applied in this study. The findings relating genetic repair deficiency to drug sensitivity (most notable under 3D conditions) are intriguing. It is also worth highlighting the negative results showing that copy number variations appear unlinked to drugs targeted to the amplified genes because this results forces us to re-examine the common assumption that overexpressed genes offer good therapeutic targets. However, without following up on the observation that homologous recombination deficiencies correlate with drug effects, there is no mechanistic insight gained from this study and the main contributions are the method and preliminary observations.

I have two major technical comments, which may affect results significantly.

1. The authors note that Hoechst and PI treatment alone causes cell death, which they correct for by normalizing all drug treatments to this untreated control. This approach unfortunately does not account for possible interactions between drug treatments and Hoechst/PI-induced cell death. A scenario where a drug induces a cell state that is more vulnerable to this Hoechst/PI-induced cell death would arise solely because of this observation-induced effect on the cells. Thus, potentially all of the observations are biased by such interactions, which could be drug or cell specific.
2. The use of a maximum intensity projection to collapse 3D imaging data for analysis may obscure phenotypes in complex 3D morphologies. For example, information from multiple cells that overlap in the projected dimension would be obscured by the brightest cell in that dimension. Such an artifact would affect the 3D conditions and not the 2D, thus inducing measurement bias across conditions.

Reviewer #2:

Summary

Failure of novel therapies in drug trials was thought to be due lack of preclinical testing on appropriate models that replicate human disease. This led to the rapid advancement of 3D models that were closer to the patient tumors than cells grown in 2D cultures. Patient tumor derived organoids are a 3D model that can be used for personalized drug screening. The advantages of organoids are the short time frame required to generate organoids, their ability to retain the stem cell population, and their ease of use for in vitro drug assays. Additionally, normal patient matched organoids can be used in drug screens to evaluate drug toxicity in normal cells. Several papers have recently been published that demonstrate the set up and use of patient derived organoids in drug screens (Francies et al., 2016; Boehnke et al, 2016).

Key conclusions

In the current study, authors have compared conventional 2D monolayer assays with organoids from patient derived xenografts cells using a novel automated microscopy based assay which they have named "DeathPro". Using this assay, they have screened nine ovarian cancer patients and shown that patient specific genomic alterations related to drug response correlated with organoids but not with 2D cultures.

Cells stained with propidium iodide and Hoechst were quantified using the "DeathPro" system and used to resolve carboplatinum induced cell death and proliferation inhibition in both ovarian and lung tumor derived organoids. Using this assay, ovarian tumor organoids and their matching 2D monolayer cells were treated with 22 drugs or their combinations. Whereas cell death in organoids was similar in 2D and 3D cultures, growth arrest varied. The authors split the drugs into "cytostatic" or "cytotoxic" categories for the purpose of analysis and found four drugs and one drug combination that exhibited lower cell death and growth arrest in organoids vs 2D cultures.

To further investigate the correlation between drug sensitivity to genomic alterations, the authors performed whole genome sequencing on patient derived xenograft cell lines. Known target genes were linked with specific inhibitor induced cell death. No correlation was observed between loss of BRCA1/2 loss or amplification of AURKA and PI3KCA and sensitivity towards DNA damage related drugs or the AURKA or PI3K inhibitor, respectively.

Importantly, the results obtained by the authors mimic patient response to drugs used in clinical trials. The authors also show that although cell death is lower in organoids 72h post-treatment, it is higher when the inhibitor is washed off, revealing an important observation that needs to be included in future studies. Finally, the heterogeneity of drug response in organoids compared to 2D cultures further establishes the use of organoids as preclinical models for in vitro drug studies.

General remarks

The paper is logical, and the conclusions are in agreement with the results obtained. The work adds to a growing field of scientists who agree on the use of organoids for drug efficacy studies. The development of the "DeathPro" assay will lead to a technical advance in the field for analysis of the efficacy of cytostatic drugs.

The advance is significant since it allows for the measurement of size of nuclei which would be time consuming to detect without the "DeathPro" assay. Scientists performing in vitro drug studies using both 2D cell cultures and organoids would be interested in this study.

Major points

- The method for generating both cell lines and organoids is unclear- it should be clarified that the organoids and cell lines are derived from patient derived xenografts, not from patient primary tumors. The materials and methods section states cells were derived directly from the tumor or first passaged through mice- it is unclear how many cell lines were directly from patients, if any, and how many were from patient derived xenografts. The references given (Noll et al. 2016 ; Aloia et al. 2015) explain the development of patient derived xenografts and 2D cultures. The media composition and conditions of growth for organoids is not clearly stated. Are these cells grown in 2D and then put into matrigel to form organoids? If so, this has an important impact on the conclusions drawn in the manuscript. The cells grown in 2D have been selected for the ability to grow as a monolayer and likely reflect a subset of aggressive tumors. These cells do not encompass the heterogeneity and the histological and molecular diversity of the tumor. It would be important to make single cell suspensions from primary patient tumors or patient derived xenografts and place directly into matrigel to form organoids.

-The use of co-cultures in 2D would add greatly to the impact of the results obtained in the manuscript. Addition of fibroblasts, preferably primary patient derived fibroblasts would add greatly to the conclusions drawn from the study. It would be important to know how organoids compare with 2D co-cultures. This arm of the study would also demonstrate the ability of the "DeathPro" assay to analyze drug induced cell toxicity and cytostatic ability in a heterogenous cell population.

Minor points

- clarification of generation of organoids
- Although the number of researchers interested in screening organoids is increasing rapidly, the authors do not mention a key drawback of using organoids- namely the lack of immune and stromal components.

Conclusions: The development of the "DeathPro" analysis is a novel and useful advancement in the field and the manuscript is a good fit for the journal. However, there are some important clarifications that need to be made and the conclusions tempered accordingly. Testing of co-cultures would add greatly to the quality of the manuscript and the experiments can be performed within 60 days. The generation of organoids directly from patient derived xenografts should increase the heterogeneity of the sample, possibly changing the conclusions drawn. If the authors can show the results are similar in a few samples (2-3), it can result in a stronger statement of the use of the "DeathPro" assay in drug testing.

Reviewer #3:

In this study "Resolving drug effects in patient-derived cancer cells links organoid responses to genome alterations", Neugebauer and her colleagues developed a new microscopy-based tool for the analysis of therapeutic response *in vitro*. Using this approach, the authors identify some differential drug sensitivity for cells grown in 2D monolayer cultures versus 3D spheroids. By correlating the genomic landscape of the models to their therapeutic responses, the authors note an association between a deficiency of DNA repair and drug sensitivity. The study presented here is very topical as 3D-cell assays are becoming more common in cancer research, and a careful comparison of 2D versus 3D culture has not been previously reported. However, the comparison provided in this paper is really 2D culture vs 2D-derived 3D cultures. A better comparison would involve primary 3D cultures (never cultured on plastic) and primary 2D cultures. Furthermore, the title and running title of this paper is confusing as the term organoid would suggest that primary patient-derived 3D cultures are used in drug screening to identify novel genetic pre-dispositions to drug sensitivity.

Overall, the manuscript is lacking in experimental details. Particularly, the drug assays that underpin the entire study need to be described in more depth. For example, the manner in which drugs were combined is not clearly stated. This is important as very effective compounds can mask the effect of weakly effective compounds. Furthermore, it would be beneficial for the authors to expand the materials and methods section about the drugs screening to clearly describe how cell death and cell growth arrest are measured (related to figure 1).

additional comments:

While the mouse fibroblast contamination is apparent in the WGS of OC22, the mouse fibroblast can also be observed in figure 2b indicating that mouse cells would be present in all cell assays. Because of the confounding effect of this cell population on the drug response, we recommend excluding OC22 from this study.

Please clearly state in the materials and methods section if the cell culture medium remains the same in the 2D and 3D culture experiments.

In figure 3a, paclitaxel appears in the wrong groups compared to 3b. Please carefully review and edit all figures for accuracy and consistency.

In Figure 3d, please clearly state why the comparison was made between 2D and 3D drug removal. Please note in the text that in 3D cultures, some therapeutic compounds will remain embedded in the Matrigel, therefore washout will never be as complete as in 2D cultures.

In figure 6 the statistical comparison of HDR and drug sensitivity does not seem robust enough to support the stated conclusions.

The addition of transcriptome analysis to this already robust data set of drug-response and whole genome analysis could uncover novel 2D vs 3D programs and could help explain the different biology in the two culture systems.

MSB-17-7697 revision

'Resolving drug effects in patient-derived cancer cells links organoid responses to genome alterations'

Neugebauer et al./Jabs et al.

(Due to marriage the family name of the first author has changed.)

First we would like to thank the editor and all reviewers for a comprehensive, constructive and thoughtful review process. The manuscript was carefully revised to address all reviewer comments, as well as the editorial requirements. Below we include a point-to-point response to illustrate changes made to the revised manuscript (Editors/Reviewer points in black font colour, authors response in blue font colour, *manuscript edits are italic*).

Editor's suggestions

The reviewer's recommendations are clear so there is no need to repeat all the points listed below. As reviewer #2 points out, the addition of further experiments demonstrating that the method works with organoids derived directly from patient tumors and patient derived xenografts and the inclusion of analyses of co-cultures would significantly enhance the impact of the study. Moreover the reviewers mention several technical issues that need to be addressed and refer to the need to describe the experimental processes and methodology in better detail.

Response to Editor

We are grateful for the editor's suggestions. In the presented, revised manuscript, we show that the *DeathPro* method works with organoids derived directly from patient-derived xenografts (Figure EV1). Additionally, we include two co-culture screens (Figure EV1, EV5, see also response 2.3) demonstrating that drug effects in 2D co-cultures resemble responses in standard 2D cultures. Furthermore, we addressed technical issues (Figure EV2, Appendix Figure 2, see also responses 1.1 and 1.2) and substantially extended the methods section to cover all experimental details. We believe that the additional experimental data and method description greatly add to the comprehensibility of our study and attractiveness of the *DeathPro* method.

Reviewer #1

Neugebauer et al. develop and implement an imaging-based method for evaluating drug effects on 2D and 3D spheroid cancer cell cultures systematically. They use this approach to show differences in drug effects on cells under different conditions and to relate these different drug actions to the genetic state of various patient derived cultures. The general field of in vitro drug evaluation and application to personalized medicine are of great interest to the medical field as well as basic scientists. Although the concept of differential drug effects in 2D vs. 3D culture is hardly novel, I am impressed by the analytical rigor applied in this study. The findings relating genetic repair deficiency to drug sensitivity (most notable under 3D conditions) are intriguing. It is also worth highlighting the negative results showing that copy number variations appear unlinked to drugs targeted to the amplified genes because this results forces us to re-examine the common assumption that overexpressed genes offer good therapeutic targets. However, without following up on the observation that homologous recombination deficiencies correlate with drug effects, there is no mechanistic insight gained from this study and the main contributions are the method and preliminary observations. I have two major technical comments, which may affect results significantly.

We would like to thank the reviewer for these positive comments.

1. The authors note that Hoechst and PI treatment alone causes cell death, which they correct for by normalizing all drug treatments to this untreated control. This approach unfortunately does not account for possible interactions between drug treatments and Hoechst/PI-induced cell death. A scenario where a drug induces a cell state that is more vulnerable to this Hoechst/PI-induced cell death would arise solely because of this observation-induced effect on the cells. Thus, potentially all of the observations are biased by such interactions, which could be drug or cell specific.

Response 1.1

We thank the reviewer for bringing this to our attention. We have addressed this concern by performing drug tests in Hoechst and PI-stained organoids as well as unstained organoids and comparing outcomes. Hoechst and PI were added to the unstained organoids only 4 h before the assay endpoint to enable imaging. We chose specifically OC12 for this experiment, as they are rather chemosensitive compared to the other OC organoids (Figure 2C, E). So far, we did not observe any additive cytotoxicity induced by the staining. We included these findings in Figure EV2 and expanded the first results section accordingly:

'Additionally, cytotoxic effects induced by 11 drugs correlated well between long-term and short-term stained organoids ($R_p=0.81-0.95$) indicating that both dyes do not interfere with drug-induced cell death measured by DeathPro (Figure EV2 C, D).'

Nevertheless, we are aware that one can never completely exclude the possibility that one of the dyes sensitize certain cells towards specific drugs or otherwise alter cellular behaviour. Still, we would like to point out that cytotoxicity in the *DeathPro* assay can be determined by endpoint staining with Hoechst and PI to limit any interference (as shown in Figure EV2 A,B). Staining at the beginning of the drug treatment is only required to accurately quantify growth for each condition, especially in 3D cultures that often suffer from heterogeneous seeding effects and growth.

2. The use of a maximum intensity projection to collapse 3D imaging data for analysis may obscure phenotypes in complex 3D morphologies. For example, information from multiple cells that overlap in the projected dimension would be obscured by the brightest cell in that dimension. Such an artifact would affect the 3D conditions and not the 2D, thus inducing measurement bias across conditions.

Response 1.2

We thank the reviewer for this constructive feedback. We are aware that our image analysis strategy is simplified which is due to the coarse image acquisition optimised for low phototoxicity and high throughput. Moreover, we focused on population-wide drug effects over several drug concentrations for the screening. Detailed confocal imaging that enables 3D image analysis is too time-consuming in drug screens, phototoxic and hard to achieve with standard instrumentation for live cells cultured on ~1mm thick Matrigel layers. The thickness of the Matrigel is however required to ensure the appropriate, tissue-like stiffness that induces organoid formation.

Although a detailed analysis of each organoid is not the focus of our work, we tested if our image analysis strategy based on maximum intensity projections (MIP) obscures cell death in organoids. For this purpose we tested it against a slice-wise analysis of confocal images. For the slice-wise analysis images were acquired with a small pinhole (pinhole <1AU, 10x objective) and at higher resolution (1.38 $\mu\text{m}/\text{pixel}$) than images collapsed into MIPs (pinhole = 200 μm , 2.67 $\mu\text{m}/\text{pixel}$). For the conditions tested, similar cell death ratios were determined with both image analysis strategies. We included these findings in Appendix Figure 1 and edited the first results section accordingly:

'To achieve low phototoxicity and high throughput of DeathPro, we chose to acquire confocal images at low resolution and to analyse 2D image projections. As 2D image analysis could bias measurements in complex 3D phenotypes, we experimentally compared the DeathPro strategy to 'slice-wise' analysis of confocal image stacks and found no difference in the cell death ratio determined by both approaches (Supplementary Fig. 3).'

Reviewer #2

Summary

Failure of novel therapies in drug trials was thought to be due lack of preclinical testing on appropriate models that replicate human disease. This led to the rapid advancement of 3D models that were closer to the patient tumors than cells grown in 2D cultures. Patient tumor derived organoids are a 3D model that can be used for personalized drug screening. The advantages of organoids are the short time frame required to generate organoids, their ability to retain the stem cell population, and their ease of use for in vitro drug assays. Additionally, normal patient matched organoids can be used in drug screens to evaluate drug toxicity in normal cells. Several papers have recently been published that demonstrate the set up and use of patient derived organoids in drug screen (Francies et al., 2016; Boehnke et al, 2016).

Key conclusions

In the current study, authors have compared conventional 2D monolayer assays with organoids from patient derived xenografts cells using a novel automated microscopy based assay which they have named 'DeathPro'. Using this assay, they have screened nine ovarian cancer patients and shown that patient specific genomic alterations related to drug response correlated with organoids but not with 2D cultures.

Cells stained with propidium iodide and Hoechst were quantified using the 'DeathPro' system and used to resolve carboplatinum induced cell death and proliferation inhibition in both ovarian and lung tumor derived organoids. Using this assay, ovarian tumor organoids and their matching 2D monolayer cells were treated with 22 drugs or their combinations. Whereas cell death in organoids was similar in 2D and 3D cultures, growth arrest varied. The authors spilt the drugs into 'cytostatic' or 'cytotoxic' categories for the purpose of analysis and found four drugs and one drug combination that exhibited lower cell death and growth arrest in organoids vs 2D cultures.

To further investigate the correlation between drug sensitivity to genomic alterations, the authors performed whole genome sequencing on patient derived xenograft cell lines. Known target genes were linked with specific inhibitor induced cell death. No correlation was observed between loss of BRCA1/2 loss or amplification of AURKA and PI3KCA and sensitivity towards DNA damage related drugs or the AURKA or PI3K inhibitor, respectively.

Importantly, the results obtained by the authors mimic patient response to drugs used in clinical trials. The authors also show that although cell death is lower in organoids 72h post-treatment, it is higher when the inhibitor is washed off, revealing an important observation that needs to be included in future studies. Finally, the heterogeneity of drug response in organoids compared to 2D cultures further establishes the use of organoids as preclinical models for in vitro drug studies.

General remarks

The paper is logical, and the conclusions are in agreement with the results obtained. The work adds to a growing field of scientists who agree on the use of organoids for drug efficacy studies. The development of the 'DeathPro' assay will lead to a technical advance in the field for analysis of the efficacy of cytostatic drugs.

The advance is significant since it allows for the measurement of size of nuclei which would be time consuming to detect without the 'DeathPro' assay. Scientists performing in vitro drug studies using both 2D cell cultures and organoids would be interested in this study.

We thank the reviewer for the very positive evaluation of our work.

The method for generating both cell lines and organoids is unclear- it should be clarified that the organoids and cell lines are derived from patient derived xenografts, not from patient primary tumors. The materials and methods section states cells were derived directly from the tumor or first passaged through mice- it is unclear how many cell lines were directly from patients, if any, and how many were from patient derived xenografts. The references given (Noll et al. 2016 ; Aloia et al. 2015) explain the development of patient derived xenografts and 2D cultures. The media composition and conditions of growth for organoids is not clearly stated.

Response 2.1

We understand that it is difficult to comprehend and evaluate the presented results without a clear description of the used cell models. Therefore, we describe now in the first methods section in detail how the patient-derived cell lines were established. We also point out which cell lines were derived from xenografts and which were directly established from the patient tumour:

'In detail, xenografts were established by first cutting primary serous adenocarcinomas into pieces < 2mm³ and then transplanting them subcutaneously into NOD.Cg-Prkdc^{scid} Il2rg^{tm1Wjl} NSG mice. Ascites or pleural effusion samples were spun down, remaining erythrocytes were removed using ACK buffer (Lonza) and the resulting cell suspension was then filtered through a 40 µm mesh (Greiner Bio-One). For initiation of xenografts, at least 1x10⁶ cells were injected intraperitoneally into NOD.Cg-Prkdc^{scid} Il2rg^{tm1Wjl} NSG mice. Mice were monitored for several months until tumor engraftment was detected. For establishment of OC PDCLs except Asc211 and PE306, engrafted tumours were taken out, cut into pieces < 1 mm³ and then enzymatically disaggregated into a single cell suspension with 1 µg/ml collagenase IV (Sigma) and DNase (Sigma) or with the human tumor dissociation kit (Miltenyi Biotec) for 2 h at 37 °C on a MACSMix rotator (Miltenyi Biotec) with occasional vortexing. Remaining erythrocytes were removed using ACK buffer. The resulting suspension was then filtered through a 40 µm mesh.

Cell lines were initiated by plating single cells suspensions (0.5 - 1 x 10⁵ cells) in T25 PRIMARIA flasks in a defined serum-free culture medium as described in (Noll et al, 2016) with the addition of 36 ng/ml hydrocortisone (Sigma), 5 µg/ml insulin (Life Technologies) and 0.5 ng/ml beta-estradiol (Sigma), referred to as CSC medium. For initial cell growth, CSC medium was supplemented with 50 µg/ml Gentamycin (Life Technologies), 0.5 µg/ml Fungizone (Life Technologies) and 10 µM ROCK inhibitor Y27632 (Selleckchem). Adherent monolayer cultures were maintained and incubated at 37 °C and 5% CO₂ and all subsequent passages were propagated without antibiotics / ROCK inhibitors. Contaminating fibroblasts

were removed by sequential differential enzymatic digestion with StemPro Accutase (ThermoFischer). Asc211 and PE306 cell lines were established directly from patient material. Cell suspensions were prepared as described above and taken directly into 2D culture. Tumorigenicity of PDCLs was verified by injecting 1×10^6 cells intraperitoneally into NOD.Cg-Prkdc^{scid} Il2rg^{tm1Wjl} NSG mice and assessment of tumour growth.'

As seen above, we include a brief description of the composition of the CSC medium, but for the sake of space, still refer to the publication of Noll et al., 2016, where the medium is described more comprehensively:

'PACO medium contains advanced Dulbecco's modified Eagle's medium- nutrient mixture F-12 (DMEM/F12; Life Technologies) with N2 supplement (Life Technologies), 50 ng/ml basic fibroblast growth factor (bFGF; Peprotech), 20 ng/ml epidermal growth factor (EGF; Peprotech), 10 ng/ml LONG R3 insulin-like growth factor-I (IGF-I) (Sigma), 100 μ M β -mercaptoethanol (Life Technologies), 2 μ g/ml heparin (Sigma).'

In addition we include a paragraph in the same methods section, which explains the growth of ovarian cancer organoids:

'To generate organoids, PDCLs and HOSEpiC were seeded onto growth-factor reduced, phenol red-free Matrigel (Corning, >9 mg/ml protein) using CSC medium supplemented with 2% (v/v) Matrigel to a density of 5000-12,500 cells cm^{-2} . Organoids were grown for up to 10 days and medium was renewed every 3-4 days to CSC medium without Matrigel.'

Are these cells grown in 2D and then put into matrigel to form organoids? If so, this has an important impact on the conclusions drawn in the manuscript. The cells grown in 2D have been selected for the ability to grow as a monolayer and likely reflect a subset of aggressive tumours. These cells do not encompass the heterogeneity and the histological and molecular diversity of the tumour. It would be important to make single cell suspensions from primary patient tumours or patient derived xenografts and place directly into matrigel to form organoids.

Response 2.2

We thank the reviewer for giving us the opportunity to explain the generation of the organoids in greater detail. The ovarian cancer organoids have not been generated independently from the cell lines but instead have been grown by seeding 2D cultured derived single cells on Matrigel. Using *DeathPro*, we decipher the influence of the extracellular matrix on cellular drug responses and try to avoid any further bias. It is not the scope of our work to compare independently generated patient models such as organoid cultures that are constantly cultured in 3D and patient-derived cell lines that are cultivated in 2D.

To enhance comprehensibility, we changed the Figure legend in Fig. 2A:

'PDCLs are maintained in 2D culture but can be grown as ovarian cancer organoids on Matrigel.'

We also adjusted the second part of the results to explicitly state that organoids are created from 2D cultured cells and to clarify our definition of a 'cancer organoid':

'PDCLs were established from metastatic serous ovarian cancers, maintained in 2D culture and seeded on Matrigel to generate 'cancer organoids' (FIGO stage IIIc-IV, Table EV1, Fig.

2A). [...] Seeded on Matrigel, HOSEpiC developed into spheres whereas PDCLs formed morphologically diverse 'cancer organoids' (Fig. 2B), that expressed the tumour markers CA-125 and WT1 (Appendix Fig. S2).'

We agree that the patient-derived cell lines cannot encompass the heterogeneity and molecular diversity of the patient tumour. Thus, like other researchers, we would optimally like to test drugs directly on heterogeneous patient material. However, a direct use comes with severe limitations regarding sample purity and amount of sample material that does not suffice for all available diagnostic procedures. In the case of OC samples as used in our study, tumour biopsies, ascites or pleural effusion samples contain a large quantity of stromal cells and immune cells that could confound the analysis of drug responses in cancer cells. Moreover, these samples harbour only very few (<1%) proliferating cancer cells. Thus, drug-induced growth arrest cannot be quantified by any biological assay as the dynamic range (<1%) is far too low. In addition, patient material is always limited in amount and often cannot be simultaneously used for functional drug testing and WGS or other diagnostic procedures. Due to low amount and scarcity of proliferating tumour cells, OC patient cells are commonly expanded *in vivo* by engrafting into mice or *in vitro* by taking cells directly into 2D culture (Ince *et al*, 2015; Janzen *et al*, 2015). The establishment of a primary OC cell line takes usually 2-6 months due to the slow growth of cancer cells and the time required for removal of stromal cells. Even organoids derived from the fast-growing, healthy colon epithelium need at least 8 weeks to be established (van de Wetering *et al*, 2015).

Consequently, generating cancer organoids directly from OC patient cells or patient-derived xenografts (PDX) is nearly impossible as cancer cells grow so slowly. In serum-containing culture, primary OC cells are viable but need 5-30 days before first passaging and become senescent after 2-5, maximum 8 passages (Mukhopadhyay *et al*, 2010). In serum-free culture, growth before the first passage also takes 10-60 days, however, cells do not become senescent within 25 passages (own experience and Ince *et al*, 2015). In addition to the slow grow-out of cancer cells, the removal of normal stromal cells from primary culture takes 4-6 passages, from human biopsy or PDX (Ince *et al*, 2015, own experience). During the establishment of the primary cell lines, numerous stromal cells but also cancer cells die. These dying cells can also seriously affect drug test results.

Regardless of these limitations and concerns, we tried to grow OC organoids directly from PDX or patient material to show the usability of *DeathPro*. In a first attempt we used frozen PDX and patient cells, as fresh patient material was not available in the required quantities. Unfortunately, nearly all cells died within 3-4 days and the few, potentially viable, PI-negative cells did not proliferate (see the Figures I, II below).

However, we succeeded in transferring OC12 cell clusters from fresh mouse ascites directly onto Matrigel and performed drug screening with *DeathPro*. The results are shown in Figure EV1 C, D and mentioned in the first results section:

'In addition, we performed pilot drug screens in OC patient cells from mouse xenografts and in 2D co-cultures with fibroblasts to validate our DeathPro concept in other common personalized cancer models (Supplementary Fig. EV1 A-D).'

Due to the long engraftment time (>3 months), we could not include other PDX. However, we argue that this cell line derived PDX serves as proof-of-principle for the usability of *DeathPro* in PDX or patient-derived cells or cell clusters.

Figure I: OC12 cells die and do not form organoids when seeded directly from PDX. OC12 cells were injected intraperitoneally into mice that consequently developed ascites. PDX cells were harvested, frozen and then seeded at indicated densities onto Matrigel. After three days cells were stained with Hoechst and Propidium iodide. Scale bar = 200 μ m. Arrows indicate large, unstained cells.

Figure II: OC patient cells die and do not form organoids when seeded from frozen patient material. Tumour was digested with Accutase, Trypsin and DNase and depleted for CD45⁺ immune cells. Cells were seeded onto Matrigel and stained at the indicated days with Hoechst and Propidium iodide. Scale bar = 20 μ m.

The use of co-cultures in 2D would add greatly to the impact of the results obtained in the manuscript. Addition of fibroblasts, preferably primary patient derived fibroblasts would add greatly to the conclusions drawn from the study. It would be important to know how organoids compare with 2D co-cultures. This arm of the study would also demonstrate the ability of the 'DeathPro' assay to analyze drug induced cell toxicity and cytostatic ability in a heterogenous cell population.

Response 2.3

We thank the reviewer for this suggestion. To prove the usability of *DeathPro* in deciphering drug effects in co-cultures we performed two drug screens with primary fibroblasts. We used human lung or ovary fibroblast to mimic cellular interactions in the metastatic niche and in the primary tumour, respectively. By staining the fibroblast with the live dye CellTracker Green, we can exclusively analyse drug effects in the cancer cells with *DeathPro*.

The co-culture principle and examples from the *DeathPro* analysis are depicted in Figure EV1 A, B and mentioned in the first results section:

'In addition, we performed pilot drug screens [...] in 2D co-cultures with fibroblasts to validate our DeathPro concept in other common personalized cancer models (Fig. EV1A-D).'

In both co-culture screens, we included the first-line therapeutics paclitaxel and carboplatin, culture-dependent drugs temsirolimus and dasatinib as well as the more effective PI3K inhibitor BKM120 to compare drug effects to standard 2D and 3D cultures. We find that drug effects are similar in both, ovary and lung co-cultures and resemble closer the effects in 2D than in 3D culture.

The results of the two co-culture screens are displayed in Figure EV5 and described in the third part of the Results section:

'As drug responses in cancer cells can be influenced by stromal cells, we investigated how drug effects change when OC cells in 2D are co-cultured with primary ovary or lung fibroblast that model cell interactions in the primary tumour or in lung metastasis, respectively. We tested four PDCLs against five OC drugs and found that drug induced cytotoxicity and growth arrest was highly correlated between both co-culture types ($R_p=0.96, 0.66$, Fig. EV5A, B). Drug responses in co-cultures resembled 2D culture effects closer than 3D culture responses ($R_p=0.64, 0.68$ vs $-0.16, -0.05$ in 2D and 3D, Fig. EV5A,B). This points to a high influence of the culture format on drug responses that even persists when the model is expanded by including other cell types.'

Minor points

- clarification of generation of organoids

Please refer to response 2.1. above.

- Although the number of researchers interested in screening organoids is increasing rapidly, the authors do not mention a key drawback of using organoids- namely the lack of immune and stromal components.

Response 2.4

We thank the reviewer for bringing this point to our attention. As suggested, we point out the importance of other cell types for realistic organoid models and drug responses and in the discussion:

'To the best of our knowledge, we provide the first detailed comparison of drug effects in primary cells cultured in the absence or presence of an extracellular matrix. By seeding patient-derived cells onto Matrigel, we generated OC organoids that, like other cancer organoids, lack stromal and immune cells as well as functional vasculature (van de Wetering et al, 2015; Schütte et al, 2017; Pauli et al, 2017). As we observed a change in drug effects upon addition of fibroblasts to 2D cultured patient cells, immune and stromal cells might also profoundly alter organoid drug responses. In the future, the OC organoid model could be enhanced by including stromal and metastases relevant cells such as mesothelial cells (Yeung et al, 2015).'

Conclusions

The development of the 'DeathPro' analysis is a novel and useful advancement in the field and the manuscript is a good fit for the journal. However, there are some important clarifications that need to be made and the conclusions tempered accordingly. Testing of co-cultures would add greatly to the quality of the manuscript and the experiments can be performed within 60 days. The generation of organoids directly from patient derived xenografts should increase the heterogeneity of the sample, possibly changing the conclusions drawn. If the authors can show the results are similar in a few samples (2-3), it can result in a stronger statement of the use of the 'DeathPro' assay in drug testing.

Response 2.5

Thank you for this positive feedback and the valuable suggestions. As described above we evaluated *DeathPro* in two different co-culture models and found that the culture format has a larger impact on drug responses than the addition of fibroblasts. In addition, we proved the applicability of *DeathPro* in cells derived directly from a PDX. By following these recommendations, we can even better showcase the power and usability of *DeathPro* for resolving drug responses in different culture models.

Reviewer #3

In this study 'Resolving drug effects in patient-derived cancer cells links organoid responses to genome alterations', Neugebauer and her colleagues developed a new microscopy-based tool for the analysis of therapeutic response in vitro. Using this approach, the authors identify some differential drug sensitivity for cells grown in 2D monolayer cultures versus 3D spheroids. By correlating the genomic landscape of the models to their therapeutic responses, the authors note an association between a deficiency of DNA repair and drug sensitivity. The study presented here is very topical as 3D-cell assays are becoming more common in cancer research, and a careful comparison of 2D versus 3D culture has not been previously reported. However, the comparison provided in this paper is really 2D culture vs 2D-derived 3D cultures. A better comparison would involve primary 3D cultures (never cultured on plastic) and primary 2D cultures.

Response 3.1

We thank the reviewer for the positive comments. Using *DeathPro*, we indeed compare 2D and 2D-derived organoid cultures to systematically decipher the influence of the extracellular matrix on cellular drug responses.

We adjusted the second part of the Results section to clearly state this aim:

'To systematically assess the influence of extracellular matrix on patient cell responses, we used the DeathPro assay to screen patient-derived OC cell lines (PDCLs) in standard 2D culture or as cancer organoids.'

Cells in primary 3D and 2D cultures independently generated from the same patient material would encounter different selection pressures. Due to various selections, both culture types would harbour different cell subtypes or cell subsets after establishment. Thus, comparing drug effects between primary cultures would likely reflect the different cell selection and culture establishment process.

As we aimed to elucidate the influence of extracellular matrix on drug responses, we chose to generate 3D cultures from 2D patient cell lines to avoid any further bias.

Furthermore, the title and running title of this paper is confusing as the term organoid would suggest that primary patient-derived 3D cultures are used in drug screening to identify novel genetic pre-dispositions to drug sensitivity.

Response 3.2

Thank you for this comment. Before deciding on 'organoid', we carefully thought about the correct term to use for our 3D cell cultures. We rejected the terms '3D culture', 'spheroid' and 'micro metastasis' for the following reasons: '3D culture' is too undefined as it refers to all different kinds of 3D cultures (matrix-based, aggregation based, solid substrate, etc.). 'Spheroid' is normally used for large cell cluster formed by matrix-free, cell aggregation-based methods like 'hanging drop' or ultra-low attachment plates. The term 'micro metastasis' would suggest the presence of other non-cancerous cell types which shape the metastatic niche and were not included in our 3D cultures.

We understand that the term 'organoid' is ambiguous due to different definitions that are used within the fields of mammary biology and intestinal biology. In the last years, the 'stem-cell' based definition of an organoid has been increasingly used. 'An organoid is now defined as a 3D structure grown from stem cells and consisting of organ-specific cell types that self-organizes ...' (Clevers, 2016). This definition typically refers to organoids deriving from stem cells of the gastrointestinal tract that are solely cultured on ECM-mimicking hydrogels. For the ovary or fallopian tube as origins of OC, no culture has been reported that complies with this stem-cell based organoid definition, probably due to the lack of stem cell markers in these epithelia.

Opposite to this definition exists an older use of the term 'organoid': 'Researchers indeed are able to generate organoids in laminin-rich gels from single cells of normal tissues or malignant tumours, or even cell lines, without necessarily starting from cells that express stem cell markers' (Simian & Bissell, 2017). As an example, non-cancerous mammary epithelial cell lines such as MCF10A or HMT-3522 S1 have been grown to breast 'acini' on Matrigel (Debnath & Brugge, 2005; Schmeichel & Bissell, 2003).

In our study, we follow this broader 'organoid' definition used in the field of gynaecological cancers. Our OC 3D cultures from 2D cultured patient cells fulfil the following important criteria for an 'organoid': They are (i) grown from single cells, on (ii) a laminin-rich hydrogel, and (iii) display morphological characteristics typical of the tissue of origin and not found in 2D culture.

Therefore, we did not change the title of our manuscript. However, we adjusted the second part of the results to explicitly state that organoids are created from 2D cultured cells and to clarify our definition of a 'cancer organoid':

'PDCLs were established from metastatic serous ovarian cancers, maintained in 2D culture and seeded on Matrigel to generate 'cancer organoids' (FIGO stage IIIc-IV, Table EV1, Fig. 2A). [...] Seeded on Matrigel, HOSEpiC developed into spheres whereas PDCLs formed morphologically diverse 'cancer organoids' (Fig. 2B), that expressed the tumour markers CA-125 and WT1 (Appendix Fig. S2).'

Nevertheless, we adjusted the running title to 'Resolving drug effects in patient cells' to point to the specific assay we developed.

Overall, the manuscript is lacking in experimental details. Particularly, the drug assays that underpin the entire study need to be described in more depth. For example, the manner in which drugs were combined is not clearly stated. This is important as very effective compounds can mask the effect of weakly effective compounds. Furthermore, it would be beneficial for the authors to expand the materials and methods section about the drugs screening to clearly describe how cell death and cell growth arrest are measured (related to figure 1).

Response 3.3

Thank you for this comment. Following your suggestion as well as the editor's recommendations, we have included all supplementary methods in the main method text.

Furthermore, we extended the 'DeathPro microscopy-based drug screens' section in the methods as follows:

'Drugs were dissolved in DMSO, water, PBS or ethanol and stored as single-use aliquots at -80°C (Table EV2). Drug dilution series (1:3) were prepared using the respective culture medium. For drug combinations, two or three drugs were combined by using similar concentrations as for single drug testing. Drug concentrations, treatment intervals and endpoints were chosen according to published studies or determined in pilot experiments. Drug screening was performed in 96-well Angiogenesis μ -Plates from ibidi. For 2D culture screens, 5,000 OC cells were seeded in 70 μ l CSC medium directly onto the plate. For organoid screens, 2,500 cells were seeded in CSC medium containing 2% Matrigel onto 10 μ l solidified Matrigel. Drugs were added in CSC medium containing 1 μ g/ml Hoechst (Invitrogen) and 1 μ g/ml PI (Sigma) one day (2D) or four days (3D) after cell seeding. After 72 h, organoids were washed twice with PBS and drug-containing medium was substituted by drug-free medium. Likewise, lung cancer cells were seeded onto Matrigel, treated with drugs in Hoechst and PI containing medium from day 4 to day 7 after seeding and incubated for another 72 h until day 10 in drug-free medium. For the 2D OC co-cultures, 1000 primary human ovary fibroblasts or IMR90 fibroblasts stained with CellTracker Green (Thermo Fischer) were seeded together with 2000 OC12 cells or 4000 cells from OC15, OC20 or PE306 PDCLs in 70 μ l CSC medium per well. Drug treatment in CSC medium started at day 1 after seeding and lasted for 72 h. In each drug assay, cells were exposed maximally to 1% DMSO or 1% ethanol. [...]

In addition, we rewrote the first methods paragraph. Now, it explains in detail the establishment of patient-derived cell lines and the growth of ovarian cancer organoids:

'[...] In detail, xenografts were established by first cutting primary serous adenocarcinomas into pieces < 2 mm³ and then transplanting them subcutaneously into NOD.Cg-Prkdc^{scid} Il2rg^{tm1Wjl} NSG mice. Ascites or pleural effusion samples were spun down, remaining erythrocytes were removed using ACK buffer (Lonza) and the resulting cell suspension was then filtered through a 40 μ m mesh (Greiner Bio-One). For initiation of xenografts, at least 1x10⁶ cells were injected intraperitoneally into NOD.Cg-Prkdc^{scid} Il2rg^{tm1Wjl} NSG mice. Mice were monitored for several months until tumour engraftment was detected. For establishment of OC PDCLs except Asc211 and PE306, engrafted tumors were taken out, cut into pieces < 1 mm³ and then enzymatically disaggregated into a single cell suspension with 1 μ g/ml collagenase IV (Sigma) and DNase (Sigma) or with the human tumor dissociation kit (Miltenyi Biotec) for 2 h at 37 °C on a MACSMix rotator (Miltenyi Biotec) with occasional vortexing. Remaining erythrocytes were removed using ACK buffer. The resulting suspension was then filtered through a 40 μ m mesh.

Cell lines were initiated by plating single cells suspensions (0.5 - 1 x 10⁵ cells) in T25 PRIMARIA flasks in a defined serum-free culture medium as described in (Noll et al, 2016) with the addition of 36 ng/ml hydrocortisone (Sigma), 5 μ g/ml insulin (Life Technologies) and 0.5 ng/ml beta-estradiol (Sigma), referred to as CSC medium. For initial cell growth, CSC medium was supplemented with 50 μ g/ml Gentamycin (Life Technologies), 0.5 μ g/ml Fungizone (Life Technologies) and 10 μ M ROCK inhibitor Y 27632 (Selleckchem). Adherent monolayer cultures were maintained and incubated at 37 °C and 5% CO₂ and all subsequent passages were propagated without antibiotics / ROCK inhibitors. Contaminating fibroblasts were removed by sequential differential enzymatic digestion with StemPro Accutase (ThermoFischer). Asc211 and PE306 cell lines were established directly from patient

material. Cell suspensions were prepared as described above and taken directly into 2D culture. Tumorigenicity of PDCLs was verified by injecting 1×10^6 cells intraperitoneally into NOD.Cg-Prkdc^{scid} Il2rg^{tm1Wjl} NSG mice and assessment of tumor growth. [...]

To generate organoids, PDCLs and HOSEpiC were seeded onto growth-factor reduced, phenol red-free Matrigel (Corning, >9 mg/ml protein) using CSC medium supplemented with 2% (v/v) Matrigel to a density of 5000-12,500 cells cm⁻². Organoids were grown for up to 10 days and medium was renewed every 3-4 days to CSC medium without Matrigel. [...]

Additional comments

While the mouse fibroblast contamination is apparent in the WGS of OC22, the mouse fibroblast can also be observed in figure 2b indicating that mouse cells would be present in all cell assays. Because of the confounding effect of this cell population on the drug response, we recommend excluding OC22 from this study.

Response 3.4

We agree that the contamination of OC22 with mouse fibroblasts poses a potential source of error. Still, we chose to include OC22 for the following reasons: Only a minority of OC22 cells (30% maximum according to purity estimation from WGS analysis) are of mouse origin and can thus only limitedly contribute to the overall OC22 drug response. Moreover, the mouse cells are present in both, 2D and 3D cultures, leading to a similar bias for the comparison of drug effects and the association with genomic data.

Please clearly state in the materials and methods section if the cell culture medium remains the same in the 2D and 3D culture experiments.

Response 3.5

We adjusted the method section accordingly:

'For drug screening, 5,000 or 2,500 OC cells were seeded in 70 μ l CSC medium directly onto the plate or on 10 μ l Matrigel for 2D and organoid culture, respectively.'

In figure 3a, paclitaxel appears in the wrong groups compared to 3b. Please carefully review and edit all figures for accuracy and consistency.

We thank the reviewer for this observation that we corrected.

In Figure 3d, please clearly state why the comparison was made between 2D and 3D drug removal. Please note in the text that in 3D cultures, some therapeutic compounds will remain embedded in the Matrigel, therefore washout will never be as complete as in 2D cultures.

Response 3.6

Thank you for this comment. In Figure 3D we try to be consistent with Figure 2E, F. As explained in the second part of the Results section, organoids grow slower than patient cells in 2D. To not generally underestimate effects in slowly-growing 3D cultures, we measured drug effects at a second time point – 72h after drug removal. We compared the results of this second time point with results from the first and only time point measured in 2D culture in Figure 2E, F as well as in Figure 3D, E.

As suggested, we adjusted the second part of Results section to point out possible effects of residual compounds in Matrigel:

‘After wash out, drug effects increased in most patient organoids (Appendix Fig. S3A-C) as they either intensify with time or continue to be induced by residual compounds in Matrigel.’

Moreover, we refined the description of drug removal in the Methods part:

‘After 72 h, organoids were washed twice with PBS and drug-containing medium was substituted by drug-free medium.’

In figure 6 the statistical comparison of HDR and drug sensitivity does not seem robust enough to support the stated conclusions.

Response 3.7

To identify correlations between HRD score and drug responses we assume a linear relationship between both variables and perform linear regression analysis. We report R^2 values to show how much variation in the data is explained by the linear model. In addition, we calculate FDRs from randomly sampled data to estimate the significance of the correlations. These FDRs correspond to p-values (e.g. FDR<0.1 for carboplatin in 2D: $R^2 = 0.62$, N=9, p-value = 0.075 or FDR<0.05 for carboplatin in 3D 72 h: $R^2 = 0.86$, N=9, p-value = 0.0029). For simplicity we only indicate correlations with an FDR<=0.1 and FDR<=0.05 and do not report all exact p-values or FDRs in Figure 6C. Out of the 20 HRD-drug response correlations we detected, six are relevant on a significance level of 0.1 and 14 are significant on a level of 0.05. As we only investigate 66 conditions in total (drug response to 22 drugs or drug combinations at three time points) these significance levels that are widely used in biological studies with low number of conditions seemed appropriate to us.

For completeness and better understanding, we included several non-significant correlations in Figure 6D-H. As this might cause misunderstandings, we specifically indicated these by changing the R^2 values to italics.

The addition of transcriptome analysis to this already robust data set of drug-response and whole genome analysis could uncover novel 2D vs 3D programs and could help explain the different biology in the two culture systems.

Response 3.8

We agree that including a transcriptome analysis to mechanistically explain the differences between both culture types would be very interesting. Nevertheless, the observed changes in drug effects could be due to a variety of altered biological processes including transcription, translation, post-translational regulation or even protein localization. Thus, transcriptome analysis would have to be complemented by various other methods. Indeed, previous protein phosphorylation studies reported a lower AKT and higher MAPK signalling activity in 3D cultured cell lines (Weigelt *et al*, 2010; Gangadhara *et al*, 2016; Luca *et al*, 2013; Riedl *et al*, 2017; Chitcholtan *et al*, 2013). This shift from AKT towards MAPK signalling could potentially explain e.g. higher efficacy of mTOR inhibitor temsirolimus in 3D compared to 2D culture. However, to conduct not only transcriptome but also proteome, protein activity and even protein localization analysis is clearly beyond the scope of our manuscript.

Final statement

In addition to the comments and suggestions provided by the editor and reviewers, we have thoroughly reviewed the manuscript to ensure correctness and made the following changes:

- In the *DeathPro* workflow (Computer Code EV1) we applied minor changes to the R-scripts to ensure reproducibility and to eliminate potential sources of errors.
- We realized that we scored HRD slightly different from the initial publication (Abkevich *et al.*, 2012) and point out the differences in the methods part.
- We exchanged the Hongisto *et al*, 2013 reference that dealt with only one breast cancer cell line to Lee *et al*, 2013 which compares responses of several OC standard cell lines cultured in 2D and 3D.
- Throughout the manuscript we introduced small changes to improve clarity.

References

- Chitcholtan K, Asselin E, Parent S, Sykes PH & Evans JJ (2013) Differences in growth properties of endometrial cancer in three dimensional (3D) culture and 2D cell monolayer. *Exp. Cell Res.* **319**: 75–87
- Clevers H (2016) Review Modeling Development and Disease with Organoids. *Cell* **165**: 1586–1597
- Debnath J & Brugge JS (2005) Modelling glandular epithelial cancers in three-dimensional cultures. *Nat. Rev. Cancer* **5**: 675–88
- Gangadhara S, Smith C, Barrett-Lee P & Hiscox S (2016) 3D culture of Her2+ breast cancer cells promotes AKT to MAPK switching and a loss of therapeutic response. *BMC Cancer* **16**: 345
- Ince TA, Sousa AD, Jones MA, Harrell JC, Agoston ES, Krohn M, Selfors LM, Liu W, Chen K, Yong M, Buchwald P, Wang B, Hale KS, Cohick E, Sergent P, Witt A, Kozhekbaeva Z, Gao S, Agoston AT, Merritt MA, et al (2015) Characterization of twenty-five ovarian tumour cell lines that phenocopy primary tumours. *Nat. Commun.* **6**: 7419
- Janzen DM, Tiourin E, Salehi JA, Paik DY, Lu J, Pellegrini M & Memarzadeh S (2015) An apoptosis-enhancing drug overcomes platinum resistance in a tumour-initiating subpopulation of ovarian cancer. *Nat. Commun.* **6**: 7956
- Luca AC, Mersch S, Schmidt S, Messner I, Schäfer K-L, Baldus SE, Huckenbeck W, Piekorz RP, Knoefel WT, Krieg A & Stoecklein NH (2013) Impact of the 3D Microenvironment on Phenotype , Gene Expression , and EGFR Inhibition of Colorectal Cancer Cell Lines. *PLoS One* **8**: e59689
- Mukhopadhyay A, Elattar A, Cerbinskaite A, Wilkinson SJ, Drew Y, Kyle S, Los G, Hostomsky Z, Edmondson RJ & Curtin NJ (2010) Development of a functional assay for homologous recombination status in primary cultures of epithelial ovarian tumor and correlation with sensitivity to poly(ADP-ribose) polymerase inhibitors. *Clin. Cancer Res.* **16**: 2344–2351
- Noll EM, Eisen C, Stenzinger A, Espinet E, Muckenhuber A, Klein C, Vogel V, Klaus B, Nadler W, Rosli C, Lutz C, Kulke M, Engelhardt J, Zickgraf FM, Espinosa O, Schlesner M, Jiang X, Kopp-Schneider A, Neuhaus P, Bahra M, et al (2016) CYP3A5 mediates basal and acquired therapy resistance in different subtypes of pancreatic ductal adenocarcinoma. *Nat Med* **22**: 278–287
- Pauli C, Hopkins BD, Prandi D, Shaw R, Fedrizzi T, Sboner A, Sailer V, Augello M, Puca L, Rosati R, McNary TJ, Churakova Y, Cheung C, Triscott J, Pisapia D, Rao R, Mosquera JM, Robinson B, Faltas BM, Emerling BE, et al (2017) Personalized In Vitro and In Vivo Cancer Models to Guide Precision Medicine. *Cancer Discov.* **7**: 462–477
- Riedl A, Schleder M, Pudelko K, Stadler M & Walter S (2017) Comparison of cancer cells cultured in 2D vs 3D reveals differences in AKT / mTOR / S6- kinase signaling and drug response. *J. Cell Sci.* **130**: 203–218
- Schmeichel KL & Bissell MJ (2003) Modeling tissue-specific signaling and organ function in three dimensions. *J. Cell Sci.* **116**: 2377–88

- Schütte M, Risch T, Abdavi-Azar N, Boehnke K, Schumacher D, Keil M, Yildiriman R, Jandrasits C, Borodina T, Amstislavskiy V, Worth CL, Schweiger C, Liebs S, Lange M, Warnatz H-J, Butcher LM, Barrett JE, Sultan M, Wierling C, Golob-Schwarzl N, et al (2017) Molecular dissection of colorectal cancer in pre-clinical models identifies biomarkers predicting sensitivity to EGFR inhibitors. *Nat. Commun.* **8**: 14262
- Simian M & Bissell MJ (2017) Organoids: A historical perspective of thinking in three dimensions. *J. Cell Biol.* **216**: 31–40
- van de Wetering M, Francies HE, Francis JM, Bounova G, Iorio F, Pronk A, van Houdt W, van Gorp J, Taylor-Weiner A, Kester L, McLaren-Douglas A, Blokker J, Jaksani S, Bartfeld S, Volckman R, van Sluis P, Li VSW, Seepo S, Sekhar Pedamallu C, Cibulskis K, et al (2015) Prospective Derivation of a Living Organoid Biobank of Colorectal Cancer Patients. *Cell* **161**: 933–945
- Weigelt B, Lo AT, Park CC, Gray JW & Bissell MJ (2010) HER2 signaling pathway activation and response of breast cancer cells to HER2-targeting agents is dependent strongly on the 3D microenvironment. *Breast Cancer Res. Treat.* **122**: 35–43
- Yeung T-L, Leung CS, Yip K-P, Au Yeung CL, Wong STC, Mok SC, Adib T, Henderson S, Perrett C, Hewitt D, Bourmpoulia D, Ledermann J, Boshoff C, Akahane T, Hirasawa A, Tsuda H, Kataoka F, Nishimura S, Tanaka H, Tominaga E, et al (2015) Cellular and molecular processes in ovarian cancer metastasis. A Review in the Theme: Cell and Molecular Processes in Cancer Metastasis. *Am. J. Physiol. Cell Physiol.* **309**: C444-56

Thank you again for sending us your revised manuscript. We have now heard back from the two referees who agreed to evaluate your study. As you will see below, the reviewers think that most of the previously raised issues have been satisfactorily addressed. However, reviewer #1 lists two remaining concerns, which we would ask you to address in a minor revision.

REVIEWER REPORTS

Reviewer #1:

I would like to thank the authors for their thorough consideration and modifications in response to the concerns I raised in the previous review. I am satisfied with the additional information provided in response to my previous point 1; however, although I appreciate the author's efforts the data provided in Appendix figure S1 do not appear to support the claim made in the text that there is "no difference in the cell death ratio determined by both approaches." In fact the data suggest to me that there may in fact be an effect due to the MIP analysis in the control sample (although there is a claim for a statistical tests, it is not clear to me what was tested). Further, the authors need to be more clear in the statistical analysis. What exactly are technical replicates? They need to specify which two means were compared using the Welch's t test as stated in the Figure legend and they need to list p values so that readers can draw their own conclusions regarding significance.

I do appreciate that it is not within the scope of this manuscript to explore the effects of 3D projecting data on their analysis pipeline, however, it must be made clear to readers that such effects are a possibility. Unless the authors are willing to perform more experiments, the authors need to note in the main text that although the MIP analysis is a necessary compromise, one should be aware of possible systemic effects. It is not the end of the world if there is an effect due to MIP analysis - my main concern is that the community ignores such a possibility. For the purposes of this publication, I would be happy if the authors remove the statement that MIP analysis results in no difference and replace it with a softer statement saying that they can still detect the effect of drug treatment and include a warning that MIP analysis may bias the results.

Reviewer #3:

The authors have responded appropriately to my questions, and the MS is much improved.

Reviewer #1

I would like to thank the authors for their thorough consideration and modifications in response to the concerns I raised in the previous review. I am satisfied with the additional information provided in response to my previous point 1; however, although I appreciate the author's efforts the data provided in Appendix figure S1 do not appear to support the claim made in the text that there is "no difference in the cell death ratio determined by both approaches." In fact the data suggest to me that there may in fact be an effect due to the MIP analysis in the control sample (although there is a claim for a statistical tests, it is not clear to me what was tested). Further, the authors need to be more clear in the statistical analysis. What exactly are technical replicates? They need to specify which two means were compared using the Welch's t test as stated in the Figure legend and they need to list p values so that readers can draw their own conclusions regarding significance.

I do appreciate that it is not within the scope of this manuscript to explore the effects of 3D projecting data on their analysis pipeline, however, it must be made clear to readers that such effects are a possibility. Unless the authors are willing to perform more experiments, the authors need to note in the main text that although the MIP analysis is a necessary compromise, one should be aware of possible systemic effects. It is not the end of the world if there is an effect due to MIP analysis - my main concern is that the community ignores such a possibility. For the purposes of this publication, I would be happy if the authors remove the statement that MIP analysis results in no difference and replace it with a softer statement saying that they can still detect the effect of drug treatment and include a warning that MIP analysis may bias the results.

Response 1

We thank the reviewer for this feedback and the reasonable concerns raised. As suggested, we softened our statement and included a warning for screening-amenable image acquisition and analysis:

"To validate that this coarse procedure captures complex 3D phenotypes, we experimentally compared the DeathPro strategy to 'slice-wise' analysis of confocal image stacks. In the tested conditions, we detected similar cell death ratios with both approaches (Appendix Fig. S1). Thus, the DeathPro imaging strategy can be used to efficiently determine drug effects in screens but at the cost of a potential bias which we cannot exclude for all conditions."

To improve the clarity of the Appendix Figure S1, we changed it as follows: We added the p-values and describe the presented data/replicates more clearly in the legend: *"Data points derive from images acquired at different positions (wells) within the same experiment."*

Final statement

In addition to the comments and suggestions provided by the editor and reviewer 1, we have made the following changes:

- In the Funding section, we specified one of our funding sources.
- We corrected small grammatical errors in the manuscript.

Corresponding Author Name: Roland Eils and Christian Conrad

Manuscript Number: MSB-17-7697